# FACT: Mitigating Inconsistent Hallucinations in LLMs via Fact-Driven Alternating Code-Text Training

**Xinxin You[1]   Qixin Sun[2]   Chenwei Yan[3]   Xiao Zhang[4]   Chen Ning[1]**
**Xiangling Fu[5]   Si Liu[2]   Guoping Hu[6]   Shijin Wang[6*]   Ji Wu[1,7*]   Xien Liu[1*]**

[1]Department of Electronic Engineering, Tsinghua University, Beijing, China
[2]School of Artifcial Intelligence, Beihang University, Beijing, China
[3]School of Information Technology and Management,
University of International Business and Economics, Beijing, China   [4]ByteDance, Beijing, China
[5]School of Computer Science, Beijing University of Posts and Telecommunications, Beijing, China
[6]iFLYTEK Research, Hefei, China   [7]College of AI, Tsinghua University, Beijing, China
{yxx23,nc22}@mails.tsinghua.edu.cn   {sunqx,liusi}@buaa.edu.cn   {chenwei.yan@uibe.edu.cn}
{zhangxiao.12@bytedance.com}   {fuxiangling@bupt.edu.cn}   {gphu,sjwang3}@iflytek.com
{wuji_ee.xeliu}@mail.tsinghua.edu.cn

## Abstract

Inconsistent hallucinations remain a major challenge for large language models (LLMs), undermining the accuracy and reliability of fact-based reasoning in real-world applications. Existing approaches often rely on task-specific training or adaptation, such as hand-crafted synthetic datasets for domain tasks or solutions mainly focused on numerical reasoning, thereby limiting generalizability to broader, unseen NLP tasks. Inspired by the structural rigor and logical consistency of programming languages, we observe that fact-based texts can be mapped to programming structures due to their inherent patterns. We further propose FACT, a novel **F**act-driven **A**lternating **C**ode-text **T**raining framework that alternates between text-to-code and code-to-text prediction. FACT is the first task-agnostic paradigm that embeds code and natural language in a shared semantic space, thereby transferring the logical consistency of code to LLM outputs in NLP tasks. Experiments show that with only a small subset of Wiki-40B-en for training, FACT reduces inconsistent hallucinations by 2.7%–8.0% and improves overall performance by 2.5%–6.1% in three leading LLMs and four diverse datasets covering QA and summarization tasks. This framework offers a new perspective on addressing challenging hallucinations in LLMs, contributing to more reliable AI.

## 1   Introduction

Achieving human-like logical consistency in fact-based reasoning is essential to advance artificial general intelligence (AGI) [1, 2]. Despite significant progress in large language models (LLMs) across various natural language processing (NLP) tasks [3, 4, 5], these models still frequently produce inconsistent hallucinations due to limited logical rigor [6, 7]. Such hallucinations occur mainly in two forms: (1) input-conflicting hallucinations, where the generated content diverges from the provided input [8, 9]; and (2) context-conflicting hallucinations, where the generated content contradicts information previously produced by the model [10, 11], as shown in the left panel of Fig. 1. These errors are particularly prevalent in fact-based reasoning, such as LLMs generating patient diagnoses in the medical domain that contradict clinical input data, or producing legal case summaries that

---

[*]Corresponding author

39th Conference on Neural Information Processing Systems (NeurIPS 2025).

misrepresent the actual case facts. Such issues severely undermine the credibility of LLMs in real-world applications. Therefore, it is urgent to develop a universal approach that can effectively address these inconsistent hallucinations across a wide range of tasks.

Existing approaches for mitigating hallucinations in LLMs often require task-specific training or adaptation. Recent efforts have addressed inconsistencies by constructing domain-specific hand-crafted reasoning datasets—for instance, LOGIQA [12] for civil service exams and AR-LSAT [13], ReClor [14] for legal reasoning. However, such manual curation is time-consuming and lacks scalability. Consequently, synthetic data approaches have emerged, aiming to automate data generation and expand coverage. Nonetheless, these methods are primarily limited to numerical reasoning tasks, such as KPDDS [15], OpenMathInstruct-1 [16], and MathGenie [17], where large-scale question-answer pair construction is tractable. Attempts to address hallucinations in broader NLP tasks include methods such as Lookback Lens and SymbCoT [18, 19]. However, they still rely on significant task-specific adaptation, including separate classifiers or symbolic CoT pipelines designed for each task. As a result, while effective in narrow domains, these approaches generally lack generalizability and scalability to diverse, unseen NLP scenarios.

Advancements in training LLMs on large-scale code datasets, such as CoCoGen and CodeRL, have significantly reduced inconsistent hallucinations in code generation tasks [20, 21]. Building on this integration of code with LLMs, recent methods further improve logical rigor in reasoning tasks. For example, program-of-thought [22] and program-assisted language models [23] translate natural language problems into executable code and utilize code interpreters to derive answers. However, these methods face practical limitations: due to fundamental differences in organizational structure, modes of expression, and language style between code and natural language, only a narrow subset of problems, typically mathematical questions, can be effectively translated into executable code [22, 23]. Consequently, these approaches fail to transfer the logical consistency benefits of code to LLM outputs in a broader range of NLP tasks.

To address the above challenges, we propose FACT: mitigating inconsistent hallucinations in LLMs via **F**act-driven **A**lternating **C**ode-text **T**raining. FACT is grounded in the observation that fact-based texts exhibit intrinsic structural patterns that are amenable to systematic mapping onto programming abstractions. As illustrated in Fig. 1 (upper right), subjects and their attributes in natural language correspond to objects and properties in code, while processes and causal relationships are represented as functions. To further transfer the logical rigor of code to natural language, FACT implements an alternating training paradigm between text-to-code and code-to-text prediction. The key challenge—the absence of ground-truth code supervision for text-to-code prediction—is addressed by employing a pseudo-labeling with a two-step verification process: 1) confirming syntactic correctness; and 2) assessing semantic fidelity to the original text. Experiments show that training on just a small Wiki-40B-en [24] subset fundamentally improves LLM output consistency. Extensive evaluations with three state-of-the-art LLMs demonstrate that FACT significantly reduces inconsistent hallucinations and improves task performance on summarization and QA tasks, all without task-specific adaptation.

We summarize the contributions of our method as follows:

- We propose FACT, the first task-agnostic framework that significantly mitigates inconsistent hallucinations in LLMs through alternating code-text training. This framework offers new perspectives for developing LLMs that deliver logical consistency and accurate responses.

- We identify a clear correspondence between the structures of fact-based text and code. This finding not only supports our alternating training methodology, but also provides a foundation for future research bridging the gap between textual and coding modalities.

- Experiments show that with only a small subset of Wiki-40B-en [24] for training, FACT reduces inconsistent hallucinations by 2.7%–8.0% and improves overall performance by 2.5%–6.1% in three leading LLMs and four diverse datasets covering QA and summarization tasks.

## 2 Related Work

**Data-Centric Approaches-Manual Bottlenecks and Narrow Scope** Recent studies suggest that hallucinations in LLMs are often attributed to deficiencies in training data, underscoring the importance of high-quality reasoning datasets for improving consistency[25, 26, 27]. Thus, domain-specific, handcrafted benchmarks have been developed, such as LOGIQA[12] for civil service exams,

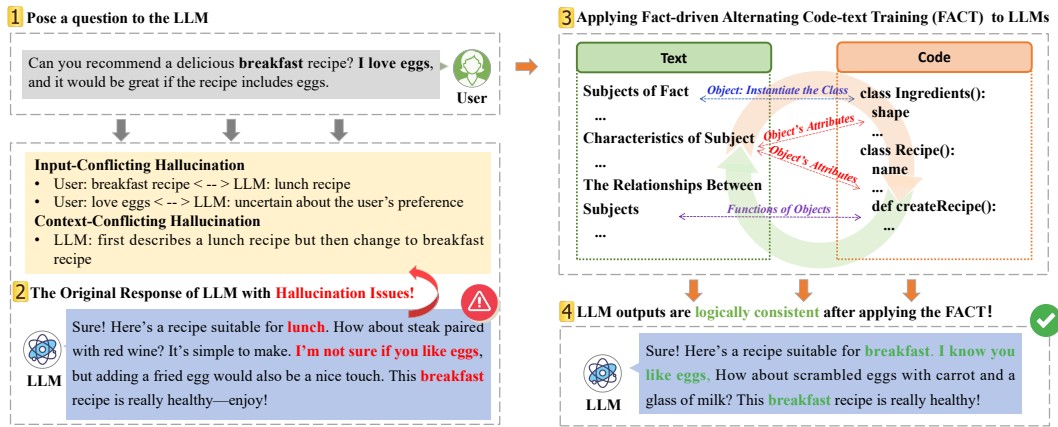

Figure 1: This figure illustrates that standard LLM responses (left) are susceptible to input-conflicting and context-conflicting hallucinations, resulting in outputs that are inconsistent with both user intent and contextual logic. The FACT approach (right) leverages structural alignment between factual text and code via fact-driven alternating code-text training, effectively reducing these hallucinations and producing responses that are both semantically and logically consistent with user intent and context.

AR-LSAT[13], and ReClor[14] for legal exams, though at considerable cost and labor. This has motivated interest in automated and synthetic dataset construction[28], particularly for numerical reasoning tasks where large-scale question-answer pairs are more easily generated (e.g., KPDDS[15], OpenMathInstruct-1[16], MathGenie[17]). However, these data-centric methods are generally tailored to specific domains or tasks, which limits their generalizability to broader NLP applications.

**Symbol-Augmented CoT Approaches-Limited by Symbolic Expressivity** Chain-of-Thought (CoT) strategies have shown promise in reducing hallucinations in reasoning tasks by promoting explicit step-by-step inference [29]. This line of work is supported by developing logical reasoning datasets such as LogiCoT [30] and FOLIO [31], which provide explicit reasoning chains. Furthermore, methods like LOGIC-LM[2] integrate symbolic solvers with language models to formalize natural language queries, while SymbCoT[19] enriches CoT prompting with symbolic expressions and logical rules. However, these methods are limited by symbolic expressivity and are less effective for complex problems, underscoring the need for more flexible and widely applicable solutions.

**Code-Based Approaches-Limited Generalization Beyond Mathematics** Recent research demonstrates that training LLMs on code datasets helps reduce hallucinations in code generation by leveraging the strict syntax and semantics of programming languages [20, 21]. Building on this integration of code with LLMs, recent methods further improve logical rigor in reasoning tasks. Approaches like Program of Thought (PoT) [22] and Program-Aided Language Models (PAL) [23] translate natural language problems into executable code and utilize code interpreters to derive answers. However, fundamental differences between code and natural language restrict these methods to mainly mathematical problems, limiting the transfer of code's logical consistency benefits to LLM outputs in a broader range of NLP tasks.

## 3 Method

Our approach builds on the observation that fact-based texts possess inherent structural properties, which can be systematically mapped to programming abstractions, as illustrated in Fig. 1 (upper right). To leverage this, we propose an alternating training paradigm between text-to-code and code-to-text generation. Due to the absence of ground-truth code for text-based code prediction, we generate pseudo-labels to supervise model training. To ensure the quality of the generated code, we apply a two-stage assessment that verifies syntactic correctness and evaluates semantic fidelity relative to the original text. This training process aligns text and code representations within a unified semantic space, effectively transferring the logical consistency of code to LLM outputs on broader NLP tasks.

The remainder of this section is organized as follows. Section 3.1 formalizes the problem setting and outlines our research objectives. Section 3.2 introduces a procedure for selecting fact-based texts.

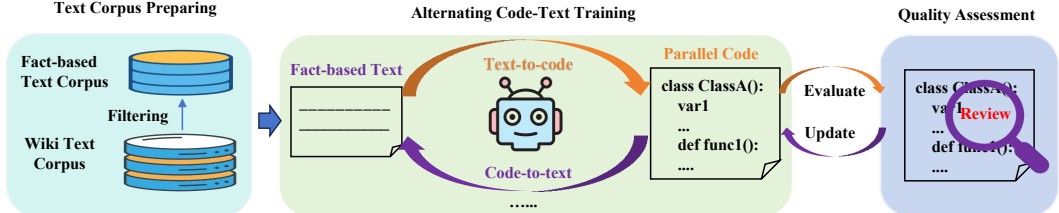

Figure 2: An overview of FACT begins with the filtering of fact-based text, followed by alternating generation training of fact-based text and parallel code based on their transformation relationship. In each iteration, a quality evaluation module is employed to assess the quality of the generated parallel code, ultimately achieving semantic and stylistic alignment between natural language and code.

Section 3.3 details the alternating code-text training mechanism. Section 3.4 describes the proposed quality assessment and adaptive loss formulation.

## 3.1 Problem Definition

We consider the task of text generation with a large language model (LLM). Given an external prompt $p$ that specifies the generation objective and an input context $t$ providing background information, the model generates an output sequence of sentences $\mathbf{x} = [x_1, x_2, \ldots, x_{|\mathbf{x}|}]$, where each $x_i$ is a sentence and $|\mathbf{x}|$ denotes the sequence length. The overall generation process can be formulated as $\mathbf{x} \sim \text{LLM}(\cdot \mid p, t)$. A central challenge in LLM-based generation is ensuring contextual and internal consistency. We focus on two types of hallucination errors that may arise in the outputs:

**Input-Conflicting Hallucination** This refers to the generated sequence $\mathbf{x}$ contains information that contradicts the input $t$. Formally, we define input-conflicting hallucination as:

$$H(\mathbf{x}, t) = \begin{cases} 1, & \exists\, \mathbf{x} \neq t \text{ and } \text{Contradict}(\mathbf{x}, t) \\ 0, & \text{otherwise} \end{cases} \tag{1}$$

**Context-Conflicting Hallucination**. This type of inconsistency arises when any two sentences within the generated output $\mathbf{x}$ are logically contradictory, even though they are generated under the same prompt and context. Formally, we define:

$$H(x_i, x_j) = \begin{cases} 1, & \exists\, i \neq j \text{ and } \text{Contradict}(x_i, x_j) \\ 0, & \text{otherwise} \end{cases} \tag{2}$$

where $\text{Contradict}(x_i, x_j)$ denotes that the $i$-th and $j$-th sentences in $\mathbf{x}$ are logically inconsistent. By formally defining these types of hallucination, our framework aims to enforce both input consistency and internal coherence in LLM-generated outputs.

## 3.2 Text Filtering for Implicit Facts

To support effective alternating training, we filter input texts to retain only those containing factual content suitable for structured code representation. This is achieved via an LLM-based filtering mechanism $E_{\text{llm}}$, which determines whether a text $t$ meets the criteria. Specifically, we use GPT-3.5 with a tailored prompt (see Appendix A.1) to perform this assessment. The decision rule is

$$D(t) = \begin{cases} 1, & \text{if } E_{\text{llm}}(t) = \text{True}, \\ 0, & \text{otherwise} \end{cases} \tag{3}$$

where $D(t) = 1$ means $t$ is selected; otherwise, it is discarded. Rigorous text filtering provides a solid foundation for the effectiveness of subsequent alternating code-text generation training.

## 3.3 Alternating Code-Text Training

Predicting code from text and reconstructing text from generated code constitute the core of our training framework, as illustrated in Fig. 2. We adopt an alternating training paradigm between

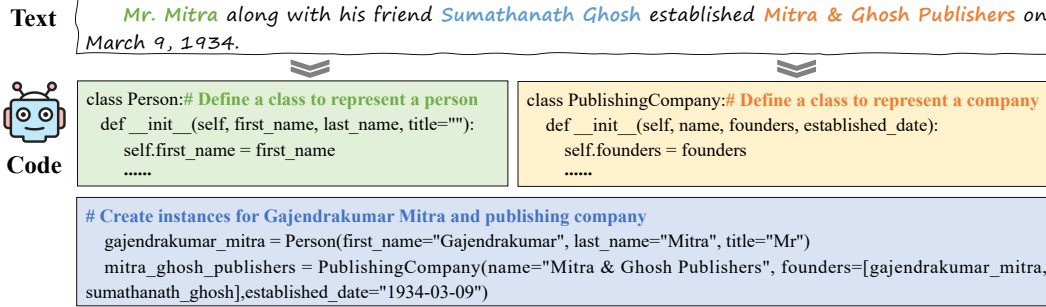

Figure 3: A simplified example showing a text segment and its corresponding code. The code first defines classes—Person and PublishingCompany—to capture the roles and structure described in the text, and then instantiates them to represent the co-founders and the company. Only a single sentence is shown here due to space constraints; another complete example is provided in Appendix C.

text-to-code and code-to-text generation, promoting the alignment of both modalities within a unified semantic space. Both directions are formulated as supervised fine-tuning (SFT) tasks.

**Text-to-Code with Pseudo-Label Supervision**   In the text-to-code direction, since ground-truth code is unavailable, we employ pseudo-label self-supervision. For each input $t$, the LLM $G$ with prompt $p_{t2c}$ (see Appendix A.2) generates a pseudo-labeled code sequence, which serves as the supervision signal for training. During training, given the same input $t$ and prompt, the model predicts the code sequence as

$$c \sim G(\cdot \mid p_{t2c}, t). \tag{4}$$

Both the pseudo-labeled code and the predicted code are produced using the same exemplar-based prompt to ensure consistency and alignment with the input semantics.

**Code-to-Text Supervision**   In the code-to-text direction, given the generated code sequence $c$ and the prompt $p_{c2t}$ (see Appendix A.3), the model reconstructs the original fact-centric text:

$$t \sim G(\cdot \mid p_{c2t}, c). \tag{5}$$

The original text is used as the ground-truth target during training. For illustration, Fig.3 presents a simplified example of both the original text and its corresponding code due to space limitations. A complete example is provided in Appendix C.

### 3.4   Quality Assessment and Adaptive Loss

While the alternating code-text training framework enables mutual supervision, it remains critical to ensure that generated code is not only syntactically correct but also semantically faithful to the source text. To address this, we introduce a two-stage quality assessment and an adaptive loss objective, explicitly incorporating both syntactic and semantic feedback into the training process.

**(1) Syntactic Validity**   Each generated code $c_i$ is executed by a Python interpreter to ensure syntactic correctness. Only code that executes successfully—i.e., free of syntax errors, undefined variables, and incomplete constructs—is retained for subsequent evaluation. For any code that fails the syntactic validity check, we assign a fixed similarity score of $S_i = 0.1$ (see Appendix B.1 for details) in the subsequent loss computation.

**(2) Semantic Fidelity via Reverse Reconstruction**   To assess whether the generated code $c_i$ captures the intended semantics of the original text $t_i$, we conduct a reverse reconstruction. With a tailored prompt $p_{\text{reorg}}$ (see Appendix A.4), let model generates a reorganized template $c_i^{\text{reorg}}$ by replacing information such as entities and attributes with code variables from $c_i$:

$$c_i^{\text{reorg}} \sim G(\cdot \mid p_{\text{reorg}}, t_i) \tag{6}$$

By executing the generated code $c_i$ and substituting runtime values into the placeholders of $c_i^{\text{reorg}}$, we obtain the instantiated text $c_i^{\text{reorg}'}$. To evaluate semantic fidelity between the generated and original

texts, we employ a composite metric combining ROUGE-1 and ROUGE-L [32], where ROUGE-1 captures factual entities, and ROUGE-L reflects global structural similarity. The overall similarity score is defined as:

$$S_i = \frac{1}{2}\left(\text{ROUGE-}L(c_i^{\text{reorg}'}, t_i) + \text{ROUGE-}1(c_i^{\text{reorg}'}, t_i)\right) \tag{7}$$

A higher value of $S_i$ indicates a greater degree of semantic fidelity of the code to the original text, as the instantiated entities and related content more closely match those described in the original. For clarity, based on the example provided in Fig. 3, we show that the reorganized text $c_i^{\text{reorg}}$ is composed of references to code entities and properties:

*mitra_ghosh_publishers.founders[0].title mitra_ghosh_publishers.founders[0].last_name*
*along with his friend mitra_ghosh_publishers.founders[1].first_name*
*mitra_ghosh_publishers.founders[1].last_name*
*established mitra_ghosh_publishers.name on mitra_ghosh_publishers.established_date.*

Upon execution of $c_i$, every placeholder in $c_i^{\text{reorg}}$ is programmatically replaced with its run-time value, resulting in the fully instantiated text $c_i^{\text{reorg}'}$ (as illustrated, all placeholders are correctly replaced):

*Mr Mitra along with his friend Sumathanath Ghosh*
*established Mitra & Ghosh Publishers on March 9, 1934.*

**Adaptive Loss** For each training example, we weight the text-to-code loss by $(1 - S_i)$, so that samples with lower quality have a larger penalty on model updates. For a batch size $n$, the final loss of the text-to-code generation process for each iteration is defined as:

$$\mathcal{L}_{\text{t2c}} = \frac{1}{n}\sum_{i=1}^{n}(1 - S_i)\,\mathcal{L}_{i,\text{t2c}} \tag{8}$$

where $\mathcal{L}_{i,\text{t2c}}$ denotes the loss of the text-to-code generation process for sample $i$ in this iteration.

In our framework, pseudo-code labels are iteratively regenerated at each training round using the latest model checkpoint, rather than being generated only once. This enables the supervision signal to dynamically improve alongside the model's evolving capabilities, yielding progressively higher-quality pseudo-labels that further enhance modality alignment. Through this alternating code-text training process, our approach establishes a robust and generalizable framework that consistently improves logical consistency and effectively mitigates inconsistent hallucinations of LLMs.

## 4 Experiment

### 4.1 Settings

**Model** We evaluated three base models: LLaMA-3.1-Instruct-8B [28], Ministral-Instruct-8B [33], and Qwen-2.5-Instruct-7B [34]. These recent and widely used instruction-tuned LLMs have comparable parameter sizes (7B/8B) and demonstrate strong performance across various NLP tasks. All models are publicly available for reproducibility.

**Datasets** We randomly sampled 10,000 entries from the Wiki-40B-en[24] dataset, using only the first paragraph of each entry. After fact-based filtering, 53.74% (5,374) were retained for alternating training, while 27.85% and 18.41% were labeled as non-factual and invalid (see 4.3). Owing to our alternating approach (text-to-code and code-to-text), each sample is used bidirectionally, yielding 10,748 samples (5,374 × 2) per iteration.

For evaluation, we selected benchmark datasets for both text summarization and question answering (QA). Specifically, we used CNN/Daily Mail[35] and SAMSum[36] for summarization, and SQuAD v2[37, 38] together with HaluEval[39] for QA. These datasets were chosen to comprehensively cover both classic and emerging scenarios: CNN/Daily Mail and SQuAD v2 are widely recognized classics, while SAMSum provides a high-quality, human-annotated benchmark for dialogue summarization, and HaluEval (introduced in 2023) specifically targets hallucination evaluation in LLMs. Since HaluEval only provides the construction methodology, we built the dataset ourselves following its

Table 1: Hallucination evaluation results for all methods and backbone models across datasets are reported. "Consis" and "AlignS" denote Consistency and AlignScore, respectively.

| Method | Summary Task | | | | QA Task | | | |
| | CNN/Daily Mail | | SAMSum | | SQuAD V2 | | HaluEval | |
| | Consis($\uparrow$) | AlignS($\uparrow$) | Consis($\uparrow$) | AlignS($\uparrow$) | Anah-v2($\downarrow$) | AlignS($\uparrow$) | Anah-v2($\downarrow$) | AlignS($\uparrow$) |
|---|---|---|---|---|---|---|---|---|
| LLaMA-3.1-Instruct-8B | | | | | | | | |
| Base | 86.40 | 83.28 | 90.79 | 90.67 | 14.32 | 95.94 | 12.48 | 96.35 |
| Prompt | 87.73 | 86.51 | 91.08 | 90.98 | 15.02 | 95.50 | 8.92 | 96.17 |
| SFT | 90.33 | 84.06 | 89.56 | 87.14 | 13.14 | 95.24 | 8.27 | 98.52 |
| SymbCoT | 86.80 | 84.28 | 90.23 | 88.37 | 13.82 | 95.79 | 11.66 | 96.58 |
| Lookback | 83.12 | 81.75 | 81.12 | 84.05 | 17.62 | 93.08 | 14.31 | 92.82 |
| FACT | **91.07** | **87.29** | **92.77** | **91.58** | **8.39** | **98.33** | **7.01** | **98.93** |
| Ministral-Instruct-8B | | | | | | | | |
| Base | 87.30 | 82.35 | 89.14 | 88.23 | 13.31 | 79.15 | 12.15 | 94.16 |
| Prompt | 88.02 | 83.12 | 90.92 | 89.10 | 14.53 | 95.64 | 13.02 | 95.11 |
| SFT | 90.10 | 83.54 | 89.45 | 87.33 | 12.65 | 86.47 | 10.36 | 96.61 |
| SymbCoT | 88.26 | 82.10 | 84.98 | 85.47 | 16.65 | 95.60 | 10.60 | 96.14 |
| Lookback | 84.37 | 80.15 | 82.13 | 85.95 | 15.40 | 92.11 | 14.26 | 93.04 |
| FACT | **90.93** | **86.21** | **93.36** | **92.63** | **9.12** | **97.75** | **7.64** | **98.84** |
| Qwen-2.5-Instruct-7B | | | | | | | | |
| Base | 83.76 | 82.56 | 85.23 | 86.42 | 17.22 | 93.65 | 14.75 | 95.05 |
| Prompt | 84.51 | 84.83 | 86.06 | 85.41 | 16.31 | 93.93 | 13.98 | 95.14 |
| SFT | 89.12 | 81.13 | 87.69 | 88.95 | 12.41 | 94.17 | 11.14 | 96.39 |
| SymbCoT | 84.45 | 81.84 | 84.31 | 86.72 | 13.25 | 93.67 | 11.38 | 95.35 |
| Lookback | 83.84 | 81.52 | 81.25 | 83.30 | 16.18 | 92.19 | 14.36 | 93.17 |
| FACT | **90.70** | **85.53** | **89.77** | **90.21** | **8.22** | **98.56** | **9.22** | **97.67** |

original paper. For fair comparison, we also randomly sampled 10,748 instances from each dataset for training the baseline methods.

**Baselines** We compare FACT with the following baselines, covering diverse modeling paradigms: (1) Base Model: the original instruction-tuned model; (2) Naive Prompting: the model was prompted with straightforward instructions (see Appendix A.5); (3) Supervised Fine-Tuning (SFT): model fine-tuned on task-specific datasets; (4) SymbCoT [19]: a recent method that reformulates chain-of-thought reasoning as symbolic inference to improve faithfulness; (5) Lookback [18]: a hallucination detection and mitigation framework using a linear classifier on lookback ratio features.

**Metrics** For hallucination evaluation, we adopt several recently proposed targeted metrics: Align-Score [40] for both summarization and QA, UniEval Consistency [41] specifically for summarization, and Anah-V2 [42] exclusively for QA. We also report overall performance metrics, including Coherence and Relevance [41] for summarization, and F1 as well as Exact Match (EM) [43] for QA.

**Implementation Details** To ensure fair comparison, all models were trained and evaluated under identical configurations. For the Base Model and Prompting variants of each backbone, inference was conducted using LLaMA-Factory[2] on four NVIDIA GeForce RTX 4090 GPUs (24GB each); both models, being instruction-tuned, adopted default chat templates and greedy decoding. FACT and SFT were also trained on the same four RTX 4090 GPUs with LLaMA-Factory for three epochs, with a learning rate of $1 \times 10^{-4}$, batch size 32, and LoRA (rank 8) for consistent acceleration. During inference, these models also employed greedy decoding. For the SymbCoT and Lookback baselines, we used the official implementations and configurations, replacing their backbone models with LLaMA, Mistral, and Qwen.

## 4.2 Evaluation Results

**Hallucination Evaluation Results** As shown in Table 1, FACT consistently outperforms all baselines across all models and datasets in hallucination metrics. For instance, with the LLaMA backbone, FACT achieves average improvements of 4.19% (Consistency) and 3.31% (AlignScore) on CNN/Daily Mail, 4.21% (Consistency) and 3.34% (AlignScore) on SAMSum, 6.39% (Anah-v2) and 3.22% (AlignScore) on SQuAD v2, and 4.12% (Anah-v2) and 2.84% (AlignScore) on HaluEval.

---

[2]https://github.com/hiyouga/LLaMA-Factory

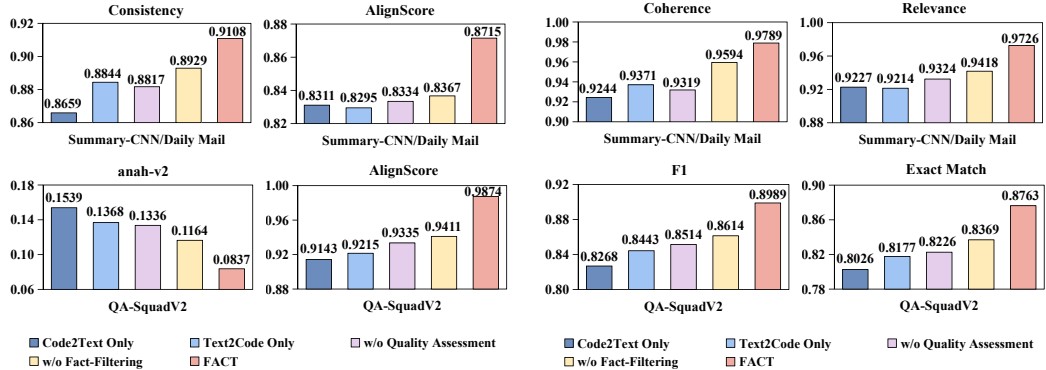

Figure 4: Ablation results on CNN/Daily Mail and SQuADv2 with the LLaMA backbone, illustrating inconsistent hallucination metrics.

Figure 5: Ablation results on CNN/Daily Mail and SQuADv2 with the LLaMA backbone, illustrating task-specific evaluation metrics.

Beyond horizontal comparisons across datasets, vertical comparisons against individual baselines on LLaMA further highlight the superiority of FACT, with average improvements of 7.57% over Lookback, 3.95% over SymbCoT, 3.49% over Base, 2.64% over SFT, and 2.57% over Prompt across all tasks. Similar results for Ministral and Qwen further highlight the robustness and generalizability of FACT. The largest margin is against Lookback, which relies on hallucination-annotated data and is thus sensitive to annotation quality, while SymbCoT, though strong for first-order logic symbolic reasoning, is less effective on tasks with complex relationships. Overall, these findings establish our fact-driven alternating code-text training as a novel framework that effectively transfers code's logical rigor to LLMs' outputs, enabling reliable and accurate reasoning across a wide range of NLP tasks.

**Task Performance Evaluation Results**    As shown in Table 2, FACT also delivers consistent improvements on both summarization and QA task metrics. For instance, with LLaMA, FACT achieves average improvements of 6.35% over Base, 3.81% over Prompt, 6.76% over SFT, 2.64% over SymbCoT, and 5.19% over Lookback across all tasks. Comparable gains are observed with Ministrel and Qwen, supporting the robustness of the conclusions. These experimental results demonstrate that enhancing consistency substantially boosts overall task performance. This improvement can be attributed to the prevention of the "hallucination snowballing" phenomenon [44], where hallucinations accumulate during generation, resulting in compounding errors and degraded output quality. By applying FACT to LLMs, we curb the accumulation of such errors, ultimately promoting the generation of more accurate outputs and thereby improving overall task performance.

**Ablation Study Results**    We conduct comprehensive ablation studies on the CNN/Daily Mail and SQuAD V2 datasets using the LLaMA backbone to systematically evaluate the contribution of each FACT module. Specifically, we examine the effects of omitting the factual text filter, disabling alternating training by using only text-to-code or only code-to-text generation, and removing the quality assessment module. **For hallucination evaluation,** as shown in Fig. 4, retaining only the code-to-text pathway results in the largest increase in hallucination rate (5.72% on average), followed by using only text-to-code (4.69%), removing the quality assessment module (4.28%), and omitting the factual text filter (3.29%). Similar patterns are observed for overall **task performance**: using only code-to-text, only text-to-code, removing the quality assessment, and omitting the factual text filter lead to mean performance drops of 6.23%, 5.13%, 4.69%, and 3.16%, respectively (see Fig. 5). These results underscore the necessity of including all modules in the FACT design. Among them, the text-to-code pathway within the alternating training scheme is particularly valuable, as it guides the model to parse and organize factual content into code-like, logically consistent representations, thereby strengthening alignment between language and structured information. In comparison, the code-to-text pathway mainly helps the model retain the ability to generate fluent and semantically coherent text.

Table 2: Task evaluation results for all methods and backbone models across datasets.

| Method | Summary Task | | | | QA Task | | | |
| | CNN/Daily Mail | | SAMSum | | SQuAD V2 | | HaluEval | |
| | Coherence | Relevance | Coherence | Relevance | F1 | Exact Match | F1 | Exact Match |
|---|---|---|---|---|---|---|---|---|
| LLaMA-3.1-Instruct-8B | | | | | | | | |
| Base | 92.40 | 92.61 | 93.25 | 90.33 | 83.59 | 79.07 | 81.33 | 77.46 |
| Prompt | 96.82 | 95.56 | 94.84 | 93.17 | 84.37 | 80.21 | 84.69 | 80.52 |
| SFT | 90.56 | 88.29 | 91.49 | 89.67 | 82.29 | 81.25 | 83.79 | 79.25 |
| SymbCoT | 96.52 | 95.47 | 96.37 | 95.78 | 82.14 | 86.20 | 85.44 | 81.73 |
| Lookback | 94.31 | 90.06 | 93.62 | 92.74 | 81.63 | 80.81 | 83.69 | 82.25 |
| FACT | **97.89** | **97.26** | **98.77** | **98.09** | **88.89** | **87.63** | **87.26** | **84.86** |
| Ministral-Instruct-8B | | | | | | | | |
| Base | 92.73 | 92.32 | 93.32 | 92.67 | 84.65 | 83.31 | 80.22 | 77.54 |
| Prompt | 94.26 | 92.87 | 94.97 | 93.65 | 85.59 | 81.78 | 82.49 | 78.20 |
| SFT | 91.04 | 90.33 | 90.25 | 90.33 | 81.77 | 80.42 | 82.86 | 81.53 |
| SymbCoT | 95.61 | 94.88 | 92.34 | 91.60 | 84.06 | 82.79 | 84.98 | 81.72 |
| Lookback | 93.32 | 92.15 | 91.29 | 90.06 | 82.44 | 80.51 | 83.65 | 82.77 |
| FACT | **96.47** | **96.16** | **96.28** | **95.39** | **88.17** | **85.36** | **86.22** | **84.13** |
| Qwen-2.5-Instruct-7B | | | | | | | | |
| Base | 89.33 | 88.45 | 88.26 | 87.34 | 80.14 | 78.25 | 78.56 | 76.47 |
| Prompt | 92.61 | 92.18 | 90.14 | 89.12 | 80.59 | 79.31 | 81.49 | 81.05 |
| SFT | 93.38 | 93.48 | 90.51 | 91.08 | 81.44 | 80.52 | 80.30 | 79.25 |
| SymbCT | 94.71 | 93.25 | 92.25 | 91.83 | 82.17 | 80.44 | 82.57 | 81.02 |
| Lookback | 94.89 | 93.62 | 91.34 | 91.58 | 80.44 | 79.16 | 81.19 | 80.44 |
| FACT | **95.50** | **95.14** | **95.72** | **94.04** | **86.59** | **84.72** | **84.34** | **83.28** |

## 4.3 Analysis and Discussion

**Analysis of the Text Filter Module**   After fact-based filtering of 10,000 Wiki-40B-en entries as described earlier, 53.74% were retained as fact-type texts, 27.85% were classified as non-fact-type, and 18.41% were labeled as invalid, largely due to unclear responses from the LLM. To further assess the reliability of the filtering process, three engineers independently annotated 100 samples each of fact-type and non-fact-type texts. Based on majority voting, the filter achieved a true positive rate of 93% and a false negative rate of 7% for fact-type texts, as well as a true negative rate of 92% and a false positive rate of 8% for non-fact-type texts. Additionally, a manual review of 100 invalid samples confirmed that 96% genuinely lacked valid labels, primarily because the model failed to follow instructions, generated responses in unexpected formats, or was unable to judge ambiguous inputs. These results demonstrate the reliability of the fact-driven filtering process, which provides a robust data foundation for subsequent alternating code-text training.

**Code Generation Quality under Pseudo-Labels**   To assess whether our pseudo-labeling with two-stage quality assessment compensates for the absence of ground-truth code, we monitor the quality of generated code across training iterations. Effective supervision should manifest as a general improvement. As shown by the purple line in the upper panel of Fig. 6, the proportion of executable code starts at 87.28%—attributable to rigorous factual text filtering—and generally increases over epochs, despite minor fluctuations, indicating enhanced syntactic validity. Similarly, semantic fidelity, measured by the similarity between reconstructed and original text, rises by up to 5.23% (red line, lower panel) before plateauing after the third iteration, which motivates our three-iterations schedule. Overall, these results show that our approach not only enables effective training without gold-standard code, but also creates a virtuous cycle in which the model's code generation ability and overall performance reinforce each other across successive iterations. Further analysis of failure cases after three iterations is provided in Appendix B.2.

**Effectiveness of Structured Code-based Inference**   To assess the effectiveness of structured code as an intermediate representation, we perform an ablation study using a two-stage inference pipeline: source texts are first converted into structured code (with unconvertible segments retained as plain text), and summaries or answers are then generated from the converted code. On the CNN/Daily Mail dataset with a LLaMA backbone, code-based inference achieves 90.24% Consistency (vs. end-to-end 91.08%, see Table 1), 86.44% AlignScore (vs. 87.15%), 97.01% Coherence (vs. 97.89%), and

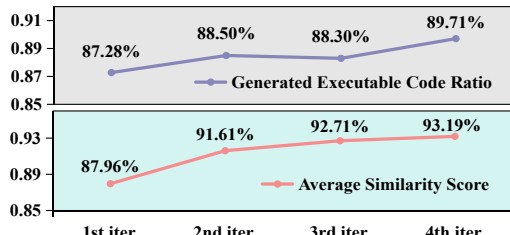
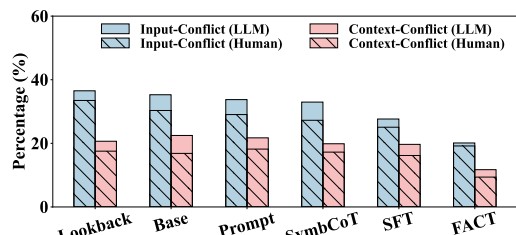

Figure 6: The trends of the proportion of generated executable code and the similarity score between reconstructed and original text as the number of training iterations increases.

Figure 7: Sentence-level hallucination rates for input-conflicting and context-conflicting hallucinations, evaluated using the GPT-4.1 and human across different methods.

96.72% Relevance (vs. 97.26%). On SQuAD V2, our pipeline yields 9.54% on Anah-v2 (vs. 8.37%), 96.81% on AlignScore (vs. 98.84%), 87.73% F1 (vs. 88.89%), and 87.05% EM (vs. 87.63%). These minimal reductions in consistency and task performance demonstrate that structured code serves as a transparent and faithful alternative representation, preserving essential semantic information for downstream tasks. This provides strong evidence for the effectiveness of the FACT framework in aligning text and code within a unified semantic space and enabling high-quality code representations.

**Analysis of Input- and Context-Conflicting Hallucinations** We evaluated FACT on challenging samples from the CNN/Daily Mail dataset, specifically selecting cases from base LLaMA outputs with consistency scores below 0.6 to target strong logical inconsistencies. Evaluation was performed both automatically, using the powerful GPT-4.1 with the prompt in Appendix A.6, and manually, by averaging the assessments of three engineers on 100 outputs randomly sampled from the above-selected cases. We quantified hallucination by measuring the proportion of sentences showing input- and context-conflicting errors. As shown in Fig. 7, FACT substantially reduces hallucination rates compared to other methods. Relative to SFT (the strongest baseline), GPT-based evaluation indicates reductions of 7.53% in input-conflicts and 5.89% in context-conflicts. Manual assessment yields similar decreases of 7.96% and 3.79%, highlighting the robustness of our findings. Additionally, context conflicts are generally less frequent than input conflicts, and FACT achieves especially notable improvements in addressing this type of hallucination. This may be attributed to the text-to-code prediction stage introduced during FACT training, where the process of generating code encourages the model to maintain higher-quality and more consistent contextual logic.

## 5  Conclusion

In this work, we present FACT, a novel, task-agnostic framework that mitigates inconsistent hallucinations in LLMs through fact-driven alternating code-text training. By unifying fact-based text and code representations in a shared semantic space, FACT brings the logical rigor and consistency of code to NLP outputs. Experiments on three leading LLMs and diverse benchmarks demonstrate that FACT greatly reduces hallucinations and consistently improves task performance. This framework not only establishes a new paradigm for addressing persistent hallucination challenges in LLMs, but also lays a solid foundation for more trustworthy and robust deployment of LLMs in real-world, fact-intensive reasoning scenarios.

## Limitations

Despite the demonstrated effectiveness of our approach, several issues remain in the generated code that warrant further attention. Some outputs still omit essential elements, such as key methods or complete attribute definitions. In addition, while overall logical consistency is generally preserved, certain complex cases exhibit incomplete handling of dependencies or lack fine-grained organization. There are also cases with class designs that do not fully represent the intended structure or relationships, as well as expressions that lack clarity or contain formatting inconsistencies. Addressing these issues—by improving content coverage, structural accuracy, and code clarity—may further enhance the robustness of FACT in future work.

## Acknowledgments

The work is supported by the Noncommunicable Chronic Diseases-National Science and Technology Major Project (Grant No.2023ZD0506501) and the Beijing Natural Science Foundation (No. 4252046). We thank the anonymous reviewers for helpful comments and feedback.

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

# Appendix and Supplemental Material

## A    Introduction to the Prompt Used

### A.1    The Prompt for Filtering Texts

*Your task is to determine whether a text is suitable for description in code (e.g., modeling a relationship using class-based instances, modeling a process using a function), and return "yes" if it is suitable, or "no" otherwise. Generally speaking, an organized text that describes a series of facts and processes is suitable for code description, while a text that mainly expresses emotions or illogicality is not suitable for code description. The following is an example:*

*Input : The Hullaballoos were created in August 1964, but had been working in the UK for over three years under the name of Ricky Knight and The Crusaders. They were not named after the American television programme Hullabaloo. Their name came from the city of Hull, England, whence they hailed.*

*Output : yes*

*Please decide whether the following text is suitable for code description according to the above requirements:*

*FK Drezga is founded on 1972, as a team from the Piperi region near Podgorica. At their first season, FK Drezga played in Fourth League - Central region (the lowest rank in SFR Yugoslavia). Club was dissolved at the end of the seventies.*

### A.2    The Prompt for Predicting Code from Text

The prompt provided to the trained base model for alternating training in predicting code from text.

*Please use a code snippet to encapsulate the given text into a structured format using classes and instances in Python. Here is an example for you.*

*Input:for Gajendrakumar Mitra: Mr Mitra along with his friend Sumathanath Ghosh established Mitra & Ghosh Publishers on March 9, 1934.*

*output:*

*#Define a class to represent a person*
*class Person:*
*    def __init__(self, first_name, last_name, title=""):*
*        self.first_name = first_name*
*        self.last_name = last_name*
*        self.title = title*
*#Define a class to represent a publishing company*
*class PublishingCompany:*
*    def __init__(self, name, founders, established_date):*
*        self.name = name*
*        self.founders = founders*
*        self.established_date = established_date*
*# Create instances for Gajendrakumar Mitra and Sumathanath Ghosh*
*gajendrakumar_mitra = Person(first_name="Gajendrakumar", last_name="Mitra", title="Mr")*
*sumathanath_ghosh = Person(first_name="Sumathanath", last_name="Ghosh")*
*# Create an instance for the publishing company Mitra & Ghosh Publishers*
*mitra_ghosh_publishers    =    PublishingCompany(    name="Mitra    &    Ghosh",    Publishers founders=[gajendrakumar_mitra,sumathanath_ghosh], established_date="1934-03-09")*

*Please complete an encapsulation of the following text based on the above requirements and particles.*

*Bacon was born in Ipswich and lived in Great Yarmouth as a child. Bacon attended Yarmouth Art School from 1917-1923, where she won a scholarship in 1917 and by 1921 passed the Board of Education drawing examinations at the earliest age possible. She studied at the Norwich School of Art and then at the Royal College of Art in London, obtaining her diploma in 1927.*

### A.3    The Prompt for Predicting Text from Code

The prompt provided to the trained base model for alternating training in predicting text from code.

*Here is a code snippet which encapsulates an original text into a structured format using classes and instances in Python. You are going to predict the original text after reading the code*

*#Define a class to represent a person*
*class Person:*
*    def __init__(self, first_name, last_name, title=""):*
*        self.first_name = first_name*
*        self.last_name = last_name*

```
        self.title = title
```
*#Define a class to represent a publishing company*
```
class PublishingCompany:
    def __init__(self, name, founders, established_date):
        self.name = name
        self.founders = founders
        self.established_date = established_date
```
*# Create instances for Gajendrakumar Mitra and Sumathanath Ghosh*
*gajendrakumar_mitra = Person(first_name="Gajendrakumar", last_name="Mitra", title="Mr")*
*sumathanath_ghosh = Person(first_name="Sumathanath", last_name="Ghosh")*
*# Create an instance for the publishing company Mitra & Ghosh Publishers*
*mitra_ghosh_publishers = PublishingCompany( name="Mitra & Ghosh", Publishers founders=[gajendrakumar_mitra,sumathanath_ghosh], established_date="1934-03-09")*

## A.4   The Prompt for Predicting Reorganized Text

The prompt provided to the trained base model to predict reorganized text based on the generated code.

*Please further utilize the original text and the generated code data to represent the entities in the original text with code, forming a new mixed text. Below is an example.*
*text=f"mitra_ghosh_publishers.founders[0].title*
*mitra_ghosh_publishers.founders[0].first_name*
*mitra_ghosh_publishers.founders[0].last_name along with his friend*
*mitra_ghosh_publishers.founders[1].first_name mitra_ghosh_publishers.founders[1].last_name established*
*mitra_ghosh_publishers.name on*
*mitra_ghosh_publishers.established_date.*

original text : The Hullaballoos were created in August 1964, but had been working in the UK for over three years under the name of Ricky Knight and The Crusaders. They were not named after the American television programme Hullabaloo. Their name came from the city of Hull, England, whence they hailed.

## A.5   The Prompt for the Base and Prompt Versions Used for Model Testing

Table 3: For the summary and QA tasks, show the instructions for the base and prompt versions of the three model bases.

| Tasks | Base | Prompt |
| --- | --- | --- |
| Summary | Write a summary of the following news. | Write a summary of the following news. Attention should be paid to the consistency of the abstract with the original text to avoid generating content with hallucinations. |
| QA | You are a question answerer. You should answer the questions directly based on the given reference without adding any prefixes or suffixes, and without analyzing the answers. After answering the question, do not say anything else. Reference document: ...... Please answer the question based on the above reference: | You are a question answerer. You should answer the questions directly based on the given reference without adding any prefixes or suffixes, and without analyzing the answers. After answering the question, do not say anything else. Please do not output content that is inconsistent with the context, and avoid giving irrelevant or contradictory answers. Reference document: ...... Please answer the question based on the above reference: |

## A.6 The Prompt for Evaluating Two Types of Inconsistent Hallucinations

Table 4: For the summary and QA tasks, provide the instructions for evaluating two types of hallucinations.

| Tasks | Input-Conflict | Context-Conflict |
|---|---|---|
| Summary | The following presents data where a large language model generates summaries based on the story content, including the original story and the model's output. Please check whether the output contains any inconsistencies with the story, such as fabricated information not present in the original story. If inconsistencies are found, return all sentences containing them; if there are none, respond with "None." | The following content is the output of a large language model. Please check for any self-contradictory parts, specifically instances where the information is inconsistent or contradicts itself. If contradictions are found, return all sentences containing them; if none are found, respond with "None." |

# B Supplement to the Experimental Section

## B.1 The setting of the $S_i$ hyperparameter.

We conducted a hyperparameter study to select the optimal penalty value for $S_i$ when the generated code is not executable. We compared several candidate values ($S_i = 0.01$, $0.1$, $0.2$, $0.3$), and for each, evaluated (1) the executable code ratio over three rounds of alternating training, and (2) the final consistency metric (hallucination rate) on the CNN/Daily Mail dataset using a Llama-based model. The results are shown in Table 5.

Table 5: Influence of penalty $S_i$ on executable code ratio (Exec. Ratio) over three alternating training rounds and consistency (all values in %).

| $S_i$ | Exec. Ratio (1st) | Exec. Ratio (2nd) | Exec. Ratio (3rd) | Consistency |
|---|---|---|---|---|
| 0.01 | 84.33% | 85.77% | 85.26% | 87.42% |
| **0.1** | **87.28%** | **88.50%** | **88.30%** | **91.08%** |
| 0.2 | 86.40% | 87.30% | 87.62% | 89.24% |
| 0.3 | 85.02% | 85.50% | 85.14% | 88.01% |

As shown in the table, $S_i = 0.1$ achieves the best trade-off in both executable code ratio and consistency. Smaller values (e.g., $S_i = 0.01$) lead to lower and less stable performance, while larger values degrade the results. Thus, we set $S_i = 0.1$ based on optimal empirical outcomes.

## B.2 Error Analysis of Low-Quality Code

We analyzed the generated code samples identified as low quality by the quality assessment module. After the third round of alternating training, we designated samples with similarity scores below a threshold of 0.8 as low quality, accounting for 13.97% of the data, while the remaining 86.03% exceeded this threshold. In addition, there were also code samples that remained non-executable after three rounds of alternating training. To gain insight into the nature of these low-quality cases, we conducted a detailed error analysis by manually evaluating a random subset of 100 non-executable samples and 100 samples with similarity scores below 0.8. This analysis revealed four main categories of errors, as illustrated in Fig. 8. These findings highlight the limitations of the current system and suggest promising directions for future research.

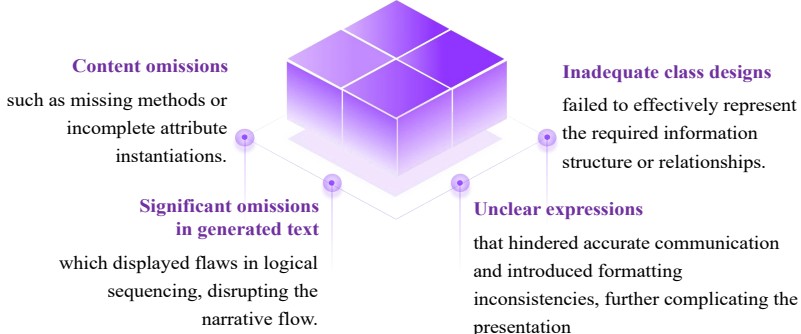

**Content omissions**
such as missing methods or incomplete attribute instantiations.

**Inadequate class designs**
failed to effectively represent the required information structure or relationships.

**Significant omissions in generated text**
which displayed flaws in logical sequencing, disrupting the narrative flow.

**Unclear expressions**
that hindered accurate communication and introduced formatting inconsistencies, further complicating the presentation

Figure 8: Classification of error causes for generated code that remained low-quality after three rounds of alternating training.

# C   A Complete Example of Text with Corresponding Generation Code

Original Text:  He was born in Santiago de Cuba, Cuba. He is Wilfredo León Hechavarría and Alina Venero Boza (a former volleyball player)'s son. He studied at Escuela Nacional del Voleibol Cubano. León Venero lived in Poland for a while with his Polish girlfriend. On July 14, 2015 he received Polish citizenship and expressed his desire to play in the Poland men's national volleyball team. On June 24, 2016 he married Małgorzata (born Gronkowska). On May 13, 2017 their daughter Natalia was born.

Corresponding Generation Code:

```
# Class representing a place with a name attribute
class Place:
    def __init__(self, name):
        self.name = name
# Class representing an education record (institution attended)
class Education:
    def __init__(self, institution):
        self.institution = institution
# Class representing a period of residence for a person at a specific location
class Residence:
    def __init__(self, person, location):
        self.person = person
        self.location = location
# Class representing citizenship information (person, country, and date obtained)
class Citizenship:
    def __init__(self, person, country, date):
        self.person = person
        self.country = country
        self.date = date
# Class representing a relationship (e.g., parent-child)
class Relationship:
    def __init__(self, person, title):
        self.person = person
        self.title = title
# Class representing a marriage
class Marriage:
    def __init__(self, person, spouse, date):
        self.person = person
        self.spouse = spouse
        self.date = date
# Class representing a child
class Child:
    def __init__(self, person, child, date):
        self.person = person
        self.child = child
        self.date = date
# Class representing a person and relevant personal relationships
class Person:
    def __init__(self, first_name, last_name, title=""):
        self.first_name = first_name
        self.last_name = last_name
        self.title = title
        self.parents = []
        self.spouse = None
        self.children = []
    #Method to add a parent object to the person's parent list
    def add_parent(self, parent):
        self.parents.append(parent)
    #Method to set spouse and marriage date
    def marry(self, spouse, date):
        self.spouse = (spouse, date)
    #Method to add a child with optional date of birth
    def add_child(self, child, date_of_birth=None):
        self.children.append((child, date_of_birth))
# Create instances for Wilfredo León Hechavarría and Alina Venero Boza
```

*wilfredo_leon_hechavarria = Person(first_name="Wilfredo", last_name="León Hechavarría")*
*alina_venero_boza = Person(first_name="Alina", last_name="Venero Boza")*
*# Create an instance for Wilfredo León Hechavarría's father relationship*
*wilfredo_leon_hechavarria_father = Relationship(person=wilfredo_leon_hechavarria, title="son")*
*# Create an instance for Wilfredo León Hechavarría's mother relationship*
*wilfredo_leon_hechavarria_mother = Relationship(person=wilfredo_leon_hechavarria, title="son")*
*# Create the main person and set parents*
*leon = Person("Wilfredo", "León Venero")*
*leon.add_parent(wilfredo_leon_hechavarria)*
*leon.add_parent(alina_venero_boza)*
*# Create an education record for the person*
*escuela = Education("Escuela Nacional del Voleibol Cubano")*
*escuela_nacional_del_voleibol_cubano = Education(institution="Escuela Nacional del Voleibol Cubano")*
*# Create a residence record for the person*
*residence = Residence(leon, "Poland")*
*wilfredo_leon_hechavarria_residence = Residence(person=wilfredo_leon_hechavarria, location="Poland")*
*# Create a citizenship record for the person*
*citizenship = Citizenship(leon, "Poland", "2015-07-14")*
*wilfredo_leon_hechavarria_citizenship = Citizenship(person=wilfredo_leon_hechavarria, country="Poland", date="2015-07-14")*
*# Create a spouse object and connect marriage to the main person*
*malgorzata = Person("Małgorzata", "Gronkowska")*
*leon.marry(malgorzata, "2016-06-24")*
*malgorzata_instance = Person(first_name="Małgorzata", last_name="Gronkowska")*
*wilfredo_leon_hechavarria_marriage =*
*Marriage(person=wilfredo_leon_hechavarria, spouse=malgorzata_instance, date="2016-06-24")*
*# Create a child object and add it to the main person with date of birth*
*natalia = Person("Natalia", "León Venero")*
*leon.add_child(natalia, "2017-05-13")*
*natalia_instance = Person(first_name="Natalia", last_name="León Venero")*
*wilfredo_leon_hechavarria_child*
*= Child(person=wilfredo_leon_hechavarria, child=natalia_instance, date="2017-05-13")*

Reorganized Text:
**Note:** For better readability, the content is displayed with appropriate line breaks.

He was born in *Place(name='Santiago de Cuba').name*, Cuba.
He is *wilfredo_leon_hechavarria.title wilfredo_leon_hechavarria.first_name*
*wilfredo_leon_hechavarria.last_name* and *alina_venero_boza.title*
*alina_venero_boza.first_name alina_venero_boza.last_name*'s *wilfredo_leon_hechavarria_father.title*.
He studied at *escuela_nacional_del_voleibol_cubano.institution*.
*wilfredo_leon_hechavarria.title wilfredo_leon_hechavarria.first_name wilfredo_leon_hechavarria.last_name*
lived in *wilfredo_leon_hechavarria_residence.location* for a while with his Polish girlfriend.
On *wilfredo_leon_hechavarria_citizenship.date*, he received Polish citizenship
and expressed his desire to play in the Poland men's national volleyball team.
On *wilfredo_leon_hechavarria_marriage.date*, he married *malgorzata_instance.title*
*malgorzata_instance.first_name malgorzata_instance.last_name*.
On *wilfredo_leon_hechavarria_child.date*, their daughter *natalia_instance.title*
*natalia_instance.first_name natalia_instance.last_name* was born.

