# OpenReview forum: "FACT: Mitigating Inconsistent Hallucinations in LLMs via Fact-Driven Alternating Code-Text Training"
_NeurIPS.cc/2025/Conference — NeurIPS 2025 poster_

### Official Review · Reviewer_PaBv · 2025-06-03

**Clarity:** 2
**Significance:** 2
**Originality:** 3
**Rating:** 4
**Confidence:** 4

**Summary:**

This paper introduces FACT, a novel training framework designed to mitigate inconsistent hallucinations in Large Language Models (LLMs). The core contribution is a training paradigm that alternates between text-to-code and code-to-text generation. The authors posit that by mapping fact-based natural language text to structured code representations (e.g., classes, objects, functions), the inherent logical consistency of programming languages can be transferred to the LLM's natural language outputs. Due to the absence of parallel ground-truth code, the method relies on generating pseudo-labels for the text-to-code task. The quality of this generated code is managed through a two-stage assessment process that checks for syntactic validity and semantic fidelity, the latter being measured by reconstructing the text from the code and comparing it to the original. The authors conduct experiments on three different LLMs across four datasets for question answering and summarization tasks, reporting that their method reduces hallucinations and improves overall task performance compared to several baselines.

**Questions:**

My assessment of this paper could improve if the authors can satisfactorily address the following fundamental questions:

1. Can you provide multiple run results for your method and key baselines (especially SFT) using different random seeds and decoding methods (not just greedy decoding)? Demonstrating results with mean, standard deviation, and appropriate statistical significance tests is crucial to proving that the reported improvements are not accidental.
2. Your method's success hinges on the initial filtering of "fact-based text" by GPT-3.5. This introduces opacity and potential bias. Could you (a) provide a formal, operational definition of "fact-based text"? And (b) provide a thorough analysis of the data that was kept versus discarded by the filter? This is crucial for understanding what your model is actually learning and for assessing the true generalizability of your approach.
3. The pseudo-labeling process is circular, and its quality is assessed using ROUGE, a metric for lexical overlap, not logical fidelity. How do you ensure the model is learning true logical relationships rather than simply learning to generate code that is lexically easy to reconstruct? Can you propose or implement a stronger, more semantically-grounded validation metric for the generated code to justify that your training signal is meaningful?

**Ethical Concerns:**

["NO or VERY MINOR ethics concerns only"]

**Final Justification:**

The author's response addressed most of the issues, but I still have concerns about the experimental results and novelty of the paper.

**Limitations:**

No. The authors include a "Limitation" section in Appendix D. However, the points discussed there (e.g., "omissions" in code, "unclear expressions") are relatively superficial implementation details.
The paper fails to acknowledge or discuss its most serious limitations, which can be referred to in Weaknesses and Questions.

**Quality:**

1

**Strengths And Weaknesses:**

This paper addresses a problem of high significance and proposes an original, creative approach. However, its contributions are undermined by severe methodological weaknesses and a lack of empirical rigor.

**Strengths:**

1. The problem of mitigating hallucinations in LLMs is one of the most critical and timely challenges in the field. Developing methods to improve the faithfulness and logical consistency of LLMs is of paramount importance for their reliable deployment in real-world applications. The paper correctly identifies this significant research gap.
2. The core idea of leveraging the structural properties of programming languages to enforce logical consistency in natural language is novel and thought-provoking. The proposed alternating code-text training paradigm is an unconventional and interesting method that moves beyond standard supervised fine-tuning or prompt engineering.
3. The paper is generally well-written, clearly structured, and easy to follow. The authors effectively use figures to illustrate their proposed framework and the problem it aims to solve.


**Weaknesses:**

1.  The central premise—that mapping factual text to code imbues it with logical consistency—is presented as an intuitive observation rather than a well-grounded theory. The paper fails to provide a formal definition of "fact-based text" or a rigorous argument explaining *why* syntactic structure should translate to semantic consistency.
2.  The methodology relies on a self-referential, circular loop of pseudo-labeling, where the model generates its own training data. This approach may reinforces the model's existing biases and flawed logical patterns, rather than correcting them. The quality assessment, which depends on a lexical overlap metric (ROUGE) for judging semantic fidelity, is superficial and insufficient to validate logical equivalence.
3. The experimental evaluation is not sound. The authors explicitly state in their checklist (Q7) that they did not report error bars or perform statistical significance testing. The reported gains could very well be statistical noise, which renders the claims of superiority over baselines unconvincing.
4. The paper repeatedly claims to be a "task-agnostic" framework. However, the evaluation is limited to summarization and question answering, two highly related tasks that often rely on extracting and reformulating factual information. There is no evidence to support the claim that this method would generalize to tasks with different structures, such as creative writing, dialogue, or reasoning. This overstatement of scope significantly diminishes the work's perceived contribution.

---

> ### Author Rebuttal · Authors · 2025-07-31
>
> We deeply appreciate your thoughtful comments and your recognition of our work. Our detailed responses to your questions are provided below.
>
> **1.Supplementary Robustness Results:** Thank you for your valuable suggestions. To address your concerns about statistical robustness, we conducted additional experiments with the LLaMA backbone on the CNN/Daily Mail (summarization) and SQuAD v2 (QA) datasets, comparing our method (FACT) to Prompt, SFT, and SymbCoT baselines. For each method (except Prompt), we ran three times with different random seeds, reporting the mean and standard deviation (in parentheses) for all hallucination metrics. As suggested, we also evaluated all methods with multiple decoding strategies: Greedy Search, Beam Search (beam=5), and top-k sampling (k=10). For the Prompt method, since it does not involve training, only the top-k decoding is affected by random seeds; therefore, we report standard deviations only for Prompt under top-k.
>
> As shown in the table below, our improvements are consistent and robust across random seeds and decoding strategies, indicating that the gains are not due to statistical noise. Furthermore, all improvements of FACT over SFT are statistically significant (paired t-test, p < 0.05).
>
> | Decoding Strategy | Methods   | CNN/Daily Mail (Consistency(↑)) | CNN/Daily Mail (AlignScore(↑)) | SQuAD V2 (Anah-v2(↓)) | SQuAD V2 (AlignScore(↑)) |
> |-------------------|-----------|----------------|---------------|------------|---------------|
> | Greedy Search     | Prompt        | 87.73          | 86.51         | 15.02      | 95.50          |
> | Greedy Search     | SymbCoT       | 86.71（0.3）    | 84.34（0.4）  | 13.61（0.4）| 95.68（0.4）  |
> | Greedy Search     | SFT           | 90.15（0.3）   | 84.20（0.3）  | 13.23（0.2）| 95.29（0.3）  |
> | **Greedy Search**     | **FACT**          | **91.11**（0.2）   | **87.27**（0.3）  | **8.41**（0.4） | **98.15**（0.4）  |
> | Beam Search       | Prompt        | 88.25          | 87.14         | 14.58      | 95.03         |
> | Beam Search       | SymbCoT       | 87.35（0.2）   | 85.47（0.2）  | 13.14（0.4）| 95.65（0.3）  |
> | Beam Search       | SFT           | 90.89（0.3）   | 84.65（0.3）  | 12.47（0.2）| 95.87（0.4）  |
> | **Beam Search**       | **FACT**          | **91.35**（0.3）   | **87.26**（0.3）  | **8.01**（0.3） | **97.54**（0.3）  |
> | top-k             | Prompt        | 87.35（0.3）   | 86.10（0.2）  | 13.84（0.2）| 95.41（0.3）  |
> | top-k             | SymbCoT       | 85.36（0.3）   | 85.24（0.4）  | 14.35（0.3）| 95.13（0.4）  |
> | top-k             | SFT           | 90.15（0.4）   | 84.05（0.3）  | 13.52（0.3）| 94.23（0.3）  |
> | **top-k**            | **FACT**          | **90.44**（0.2）   | **86.54**（0.4）  | **8.33**（0.3） | **96.25**（0.2）  |
>
> **2.Fact-based Text: Definition and Analysis:** Thank you for your valuable comment. In our work, we define a fact-based text as a passage that expresses objective, verifiable information about entities, their attributes, relationships, or processes, which can be concretely mapped to structured code constructs (e.g., class instances, properties, or procedural functions). This aligns with Iso et al. (2020), who describe fact-based texts as those that “describe a series of facts and processes in an organized manner”.
>
> > Reference: Iso, H., Qiao, C., & Li, H. (2020). Fact-based Text Editing. Proceedings of the 58th Annual Meeting of the Association for Computational Linguistics, 171–182.
>
> To address the potential opacity and bias introduced by the fact-based text filter, we summarized the characteristics of the retained fact-based texts and the discarded non-factual texts, as shown in the table below. Specifically:
>
> 1. **Quantitative analysis:** The retained fact-based texts have much higher average entity and relation counts, as automatically extracted by DeepSeek-R1 and then manually checked (93% accuracy on 100 random samples to ensure reliability), reflecting richer and more structured information compared to the discarded non-factual texts.
>
> 2. **Topic patterns:** We used DeepSeek-R1 for automatic topic analysis and manually verified 100 random cases, achieving 91% accuracy to ensure reliability. We then summarized the topic characteristics based on the top 30 frequent topics. These results show that the filter effectively separates objective, factual content from subjective or unverifiable information.
>
> 3. **Representative examples:** These further highlight this distinction: fact-based samples focus on concrete, verifiable information, while discarded samples are characterized by subjectivity or ambiguity.
>
> | Text Type                   | Avg. Entities          | Avg. Relations        | Topic Analysis                                                                                                 | Example Snippet                                                                                                        |
> |-----------------------------|---------------|----------------|---------------------------------------------------------------------------------------------------------------|-----------------------------------------------------------------------------------------------------------------|
> | Retained Fact-based Texts    | 11.3                | 7.4                  | Objective descriptions: company information, product attributes, competitions, organizational situations, etc. | ...Ger O'Brien represented the Eircom League International squad in Aberdeen for the quadrangular tournament with England, Scotland and Wales in 2004... |
> | Discarded Non-factual Texts  | 4.3                  | 2.2                   | Subjective or imaginative content: reviews, personal opinions, hypothetical scenarios, unverifiable statements, or frequent hedging expressions. | ...Commenters on social media pointed out that the number could also be seen as a reference to a joke in the film Anchorman... |
>
> While our definition of fact-based text is operational rather than theoretically rigorous, the above analysis shows that this filtering reliably selects comprehensive, objective content suitable for structured code mapping. Prior work (e.g., CoCoGen, CodeRL) demonstrates that code-based training reduces inconsistencies in LLM outputs on code generation tasks. FACT further leverages the structural alignment between factual text and code through alternating code-text training, thereby transferring this logical consistency to LLM outputs on NLP tasks.
>
> **3.Evaluation of Generated Code:** Thank you for raising this important issue. We chose the combination of ROUGE-1 and ROUGE-L as our semantic fidelity metrics due to their complementary strengths: ROUGE-1 captures factual tokens and entities, while ROUGE-L reflects global word order and structural similarity.
>
> Additionally, we evaluated the quality of the generated code across training iterations using BERTScore, which measures deeper semantic fidelity through contextual embeddings. As shown in the table below, $S_i$ (ROUGE-based metric as defined in Eq. 7 of the main paper) and BERTScore remain closely aligned and both improve across training rounds.
>
> |  Iteration  | $S_i$  Score (%) | BERTScore (%) |
> |:-----------:|:-----------:|:-------------:|
> |  Round 1    |   87.96     |     88.45     |
> |  Round 2    |   91.61     |     91.82     |
> |  Round 3    |   92.71     |     92.50     |
>
> Moreover, assessing true logical equivalence between reconstructed text based on generated code and original texts is challenging. To address this, we performed a deeper evaluation using DeepSeek-R1, splitting data after the third training round into High-Quality (Si > 0.8) and Low-Quality (Si ≤ 0.8) subsets. We measured entity and relation alignment rates, both extracted by DeepSeek-R1 and manually counted. To ensure reliability, manual verification of 100 random samples showed 92% overall accuracy for DeepSeek-R1. As shown in the table below, the High-Quality set exhibits much stronger alignment, indicating that iterative pseudo-labeling enables the model to capture logical relationships beyond surface similarity. These findings support that ROUGE-based evaluation, though simple, aligns well with more rigorous metrics and serves as a practical proxy for semantic fidelity.
>
> |   Dataset    | Entity Alignment Rate (%) | Relationship Alignment Rate (%) |
> |:------------:|:----------------------------------:|:-----------------------------:|
> | **High-Quality** |             **92.35**                 |            **86.26**              |
> | Low-Quality  |             54.33                 |            38.16              |
>
> **4.Discussion on Task-Agnostic Generalization:** Thank you for your insightful comments. We chose summarization and question answering because they are common tasks where LLMs frequently hallucinate and have well-established benchmarks [1]. Moreover, these tasks are not solely focused on factual extraction and reformulation. For example, SAMSum is a dialogue-based summarization dataset that requires models to understand conversational context, while SQuAD v2 includes a broad range of answer types (e.g., dates, numbers, entities) and demands diverse reasoning abilities such as syntactic and lexical variation, multi-sentence reasoning, and ambiguity resolution.
>
> By "task-agnostic," we mean our framework does not require dataset-specific training or adaptation and can be directly applied to different tasks. We acknowledge that broader evaluation on more text generation tasks is needed to fully demonstrate task-agnosticism. We will clarify this and use more precise language in the revised manuscript.
>
> > [1] Huang L, Yu W, Ma W, et al. A survey on hallucination in large language models: Principles, taxonomy, challenges, and open questions. ACM Transactions on Information Systems, 2025, 43(2): 1-55.

---

> > ### Comment · Reviewer_PaBv · 2025-08-01
> > **Follow up**
> >
> > Thanks for the author's reply. Based on your reply, I have some additional questions.
> >
> > The experiments are still insufficient. Three repeated experiments are not enough for LLMs, partly because random sampling greatly affects performance, and also because three experiments can introduce bias into statistical tests. The experimental dataset is also not sufficient, and there are few baselines. In many settings, performance improvements are within the margin of error. The authors also did not provide details of the statistical tests. Therefore, I remain skeptical of the experimental results.
> >
> > I still believe that the use of ROUGE here fails to achieve the author's intended purpose. Furthermore, BERTScore, as a semantic similarity metric, cannot involve logic and other factors. The authors should attempt to explain from a theoretical (mathematical) perspective why these metrics are feasible, and provide some examples to enhance understanding. This is a critical design in the paper, which directly affects the results and requires more thorough discussion.
> >
> > Furthermore, I believe the article's novelty is also low. Especially the idea of utilizing code is already very common, for example [1], but the author did not even discuss it.
> >
> > Given the weak evaluation, unclear method choices, and low novelty, I am still inclined to reject this paper. Therefore, I will maintain my current score, but I am willing to reconsider if the authors can effectively address the issues I mentioned above.
> >
> > [1] Self-play with Execution Feedback: Improving Instruction-following Capabilities of Large Language Models; ICLR 25

---

> ### Author Response · Authors · 2025-08-06
> **The issue of experiments （1/2）**
>
> Thank you for the time and effort you have devoted to reviewing our submission. Your continued feedback has been extremely valuable in helping us improve the quality of the manuscript. We sincerely value your concerns and are committed to engaging with these issues thoughtfully. We hope our detailed responses will provide the necessary clarification.
>
> We fully understand your concern regarding the potential bias in performance evaluation caused by random sampling, as well as your expectation for additional repeated experiments. Therefore, building upon the three repeated experiments included in our initial response — which followed a common practice observed in many LLM research papers — we further increased the number of repetitions to ensure that all core experimental comparisons were conducted five times. We report the results as the **mean and standard deviation** across these runs for all hallucination metrics presented in **Table 1**.
>
> Table 1:  Comparison results of mean (standard deviation) over five repeated times.
>
> | Decoding Strategy | Methods    | CNN/Daily Mail (Consistency(↑)) | CNN/Daily Mail (AlignScore(↑)) | SQuAD V2 (Anah-v2(↓)) | SQuAD V2 (AlignScore(↑)) |
> |-------------------|------------|----------------------------------|-------------------------------|------------------------|--------------------------|
> | Greedy Search     | Base       | 86.40                           | 83.28                         | 14.32                  | 95.94                    |
> | Greedy Search     | Lookback   | 83.05 (0.2)                      | 81.70 (0.3)                    | 17.49 (0.3)            | 93.09 (0.2)              |
> | Greedy Search     | Prompt     | 87.73                            | 86.51                         | 15.02                  | 95.50                    |
> | Greedy Search     | SymbCoT    | 86.52 (0.3)                      | 84.33 (0.3)                   | 13.70 (0.2)             | 95.57 (0.2)              |
> | Greedy Search     | SFT        | 90.11 (0.3)                      | 84.20 (0.4)                    | 13.34 (0.3)            | 95.24 (0.3)              |
> |**Greedy Search**     | **FACT**       | **91.07** (0.2)                      | **87.29** (0.2)                   | **8.39** (0.3)             | **98.33** (0.2)              |
> | Beam Search       | Base       | 86.56                            | 83.58                         | 13.51                  | 96.22                    |
> | Beam Search       | Lookback   | 83.17 (0.2)                      | 81.77 (0.2)                   | 17.37 (0.2)            | 93.16 (0.3)              |
> | Beam Search       | Prompt     | 88.25                           | 87.14                         | 14.58                  | 95.03                    |
> | Beam Search       | SymbCoT    | 87.20 (0.3)                      | 85.66 (0.3)                   | 13.07 (0.3)            | 95.43 (0.2)              |
> | Beam Search       | SFT        | 90.81 (0.3)                      | 84.57 (0.4)                   | 12.54 (0.4)            | 95.88 (0.3)              |
> | **Beam Search**       | **FACT**       | **91.55** (0.2)                      | **87.51** (0.2)                   | **8.06** (0.3)             | **97.42** (0.2)              |
> | top-k             | Base       | 86.14 (0.2)                       | 83.10 (0.3)                   | 13.66 (0.3)             | 95.45 (0.3)               |
> | top-k             | Lookback   | 83.00 (0.3)                       | 81.66 (0.2)                   | 17.41 (0.4)            | 92.94 (0.2)              |
> | top-k             | Prompt     | 87.39 (0.4)                      | 86.00 (0.2)                    | 13.88(0.2)            | 95.43(0.3)              |
> | top-k             | SymbCoT    | 85.27 (0.4)                      | 85.23 (0.2)                   | 14.43 (0.3)            | 95.27 (0.3)              |
> | top-k             | SFT        | 90.06 (0.3)                      | 84.01 (0.2)                   | 13.47 (0.2)             | 94.13 (0.3)              |
> | **top-k**             | **FACT**       | **90.58** (0.2)                      | **86.74** (0.3)                   | **8.30** (0.3)              | **96.10** (0.2)               |

---

> ### Author Response · Authors · 2025-08-06
> **The issue of experiments （2/2）**
>
> Based on the results of five runs, the performance improvements of FACT over all baselines are statistically significant (**p < 0.05**). The details of the paired t-tests comparing our method (FACT) with all baselines are presented in **Table 2**. These results demonstrate that the observed performance gains are consistent and not attributable to statistical noise or random fluctuations. We sincerely hope for your understanding, as conducting a large number of repeated comparative experiments on LLM-based methods within a short period is extremely challenging for academic research institutions due to the high computational demands. In summary, we are deeply grateful for your thorough and meticulous evaluation, which has significantly contributed to improving the completeness of our work.
>
> Table 2: The results of paired t-tests comparing FACT with all baselines for which multiple runs were available.
>
> | Decoding Strategy | Comparison methods       | CNN/Daily Mail (Consistency) | CNN/Daily Mail (AlignScore) | SQuAD V2 (Anah-v2) | SQuAD V2 (AlignScore) |
> |-------------------|-------------------------|-----------------------|------------------------|-------------------|---------------------|
> | Greedy Search     | FACT vs. Lookback       | 3.10E-06              | 8.00E-06               | 6.00E-07          | 5.80E-06            |
> | Greedy Search     | FACT vs. SymbCoT        | 3.17E-05              | 1.41E-04               | 1.00E-07          | 2.83E-05            |
> | Greedy Search     | FACT vs. SFT            | 2.36E-03              | 5.30E-06               | 7.00E-06          | 8.90E-06           |
> | Beam Search       | FACT vs. Lookback       | 3.00E-07              | 3.00E-07               | 4.00E-06         | 3.30E-06            |
> | Beam Search       | FACT vs. SymbCoT        | 6.90E-05              | 1.55E-03               | 2.04E-05          | 3.78E-04            |
> | Beam Search       | FACT vs. SFT            | 2.14E-02              | 3.64E-04               | 1.09E-04          | 2.00E-03            |
> | top-k             | FACT vs. Base           | 2.29E-05              | 1.90E-04               | 1.09E-05          | 2.82E-02            |
> | top-k             | FACT vs. Lookback       | 8.00E-07              | 1.42E-05               | 9.00E-07          | 6.35E-05            |
> | top-k             | FACT vs. Prompt         | 3.90E-05              | 3.14E-02              | 1.04E-05          | 3.64E-02            |
> | top-k             | FACT vs. SymbCoT        | 1.10E-06              | 2.02E-03               | 8.00E-06          | 7.65E-03            |
> | top-k             | FACT vs. SFT            | 3.48E-02              | 1.14E-04               | 6.00E-07          | 5.18E-05            |

---

> ### Author Response · Authors · 2025-08-06
> **The issue of the ROUGE metric**
>
> We deeply appreciate your rigor and sincerely thank you for your valuable suggestions. We also apologize for not clearly articulating these points in our initial response. In this round of response, we will elaborate on this issue from the perspective of the overall framework design, and we sincerely hope this will help address your concerns and reservations.
>
> As part of our Fact-Driven Alternating Code-Text Training framework, we evaluate the quality of generated code through two complementary strategies: (1) assessing its executability, and (2) measuring factual consistency by comparing the reconstructed factual text with the original. In our manuscript, we employ a ROUGE-based metric—a combination of ROUGE-1 and ROUGE-L, as discussed in our initial response, since they offer complementary granularity—to assess the consistency between the reconstructed text and the original factual text. The ROUGE-based metric is used for its simplicity and demonstrated effectiveness in our experiments. It is also highly convenient to incorporate this metric into the algorithm’s loss function for text-code iterative training, continuously improving the quality of fact-based code generation from text. Additionally, as stated in our initial response, we further incorporated BERTScore (which measures semantic fidelity through contextual embeddings) to validate the evaluation of the generated code quality across training iterations. Table 3 illustrates that: (1) the quality of generated code improves throughout the iterative training process, highlighting the effectiveness of integrating quality metrics into the loss function; and (2) the ROUGE-based scores are largely consistent with BERTScore, supporting the reliability of the evaluation results.
>
> Table 3: Semantic fidelity evaluation of the original and reconstructed texts at each iteration.
>
> | **Iteration** | **ROUGE-based Score $S_i$ (%)** | **BERTScore (%)** |
> |:-------------:|:---------------------------:|:-----------------:|
> | Round 1       | 87.96                       | 88.45             |
> | Round 2       | 91.61                       | 91.82             |
> | Round 3       | 92.71                       | 92.50             |
>
> Inspired by your suggestion as well as that of another reviewer, we further introduced an alignment-based analysis during the response phase—specifically, evaluating the entity and relationship alignment rate between the reconstructed text and the original factual text—to further assess the effectiveness of the ROUGE-based metric. Specifically, we divided the reconstructed texts based on their ROUGE-based scores $S_i$ into two subsets: High-Quality ($S_i$>0.8) and Low-Quality ($S_i$≤0.8). We then measured the alignment of entities and relationships between the reconstructed texts and the original factual texts within each subset. If a reconstructed text is fully aligned with the original in terms of entities and relationships, it indicates that the facts described in the reconstructed text are consistent with those in the original. The results in Table 4 indicate that the high-quality subset, as measured by the ROUGE-based metric, exhibits significantly higher alignment rates than the low-quality subset. This supports the reasonableness and effectiveness of using the ROUGE-based metric for assessing factual consistency.
>
> Table 4: Alignment rate comparison between high-quality and low-quality subsets.
>
> | **Dataset**    | **Entity Alignment Rate (%)** | **Relationship Alignment Rate (%)** |
> |:--------------:|:----------------------------:|:-----------------------------------:|
> | **High-Quality**   | **92.35**                        | **86.26**                              |
> | Low-Quality    | 54.33                        | 38.16                              |
>
> We fully understand and appreciate your insistence on a solid theoretical foundation for evaluation metrics. However, achieving such theoretical guarantees is often extremely challenging—and in most practical scenarios, not always feasible. Overall, although the ROUGE-based metric is not a theoretically perfect evaluation standard, our supplementary analyses using BERTScore and the entity and relationship alignment rate demonstrate that, from both practical and empirical perspectives, it serves as a simple yet effective metric that contributes positively to the iterative training process of our method. Finally, we sincerely thank you for your insightful suggestions and thoughtful concerns, which have prompted us to engage in challenging yet meaningful discussions and reflections that have ultimately strengthened our work.

---

> ### Author Response · Authors · 2025-08-06
> **The issue of discussion of other work about using code**
>
> We sincerely thank you for providing this meaningful reference. It is indeed an interesting paper. The proposed AUTOIF[1] method partially transforms LLM instructions into executable code (e.g., converting instructions like "limit the response to N words" into verifiable functions) to evaluate the instruction-following capability of LLMs. **In contrast, our proposed submission, FACT, is grounded in the observation that fact-based texts can be systematically mapped to programming structures.** It introduces an innovative fact-driven alternating code-text training framework that alternates between text-to-code and code-to-text prediction during training. This bidirectional process enables the consistency inherent in code to be transferred to the outputs of language models across various NLP tasks, thereby effectively mitigating inconsistent hallucinations.
>
> The main differences between the two methods are:
>
> **1) Objective:** AUTOIF focuses on automatically generating high-quality instruction-following training data, whereas FACT is designed to reduce inconsistent hallucinations in LLM outputs;
>
> **2) Role of Code:** AUTOIF treats code as an auxiliary tool for data filtering only, without incorporating code into the model’s training process. In contrast, FACT regards code as an alternative, structured representation of factual information and integrates code directly into model training. By alternating between code and text during training, FACT enables deep logical understanding and semantic alignment at the structural level, allowing the logical consistency of code to be effectively transferred to LLM outputs across various NLP tasks.
>
> Overall, in terms of motivation, objectives, framework design, and experimental setting, these are two fundamentally different works. That said, we fully agree with your suggestion that both are related to code, and we will include a discussion of this related work in the revised version.
>
> [1] Self-play with Execution Feedback: Improving Instruction-following Capabilities of Large Language Models; ICLR 25

---

> > ### Comment · Reviewer_PaBv · 2025-08-07
> >
> > Thank you for the detailed response; most of my concerns have been resolved. I hope the authors can include more details in the revised version. However, I still believe the experimental results and the novelty of the paper are insufficient. Therefore, I will raise my score to weak accept.

---

> > > ### Author Response · Authors · 2025-08-07
> > >
> > > Thank you very much for your valuable comments and the time you dedicated to reviewing our manuscript. We greatly appreciate your suggestions, and we will incorporate these points in the revised manuscript.

---

### Official Review · Reviewer_mHXd · 2025-06-12

**Clarity:** 3
**Significance:** 4
**Originality:** 4
**Rating:** 6
**Confidence:** 4

**Summary:**

This paper proposes a training method to reduce hallucinations in LLMs. The method FACT involves an alternating training paradigm between text-to-code and code-to-text. This is motivated by the fact that fact-based text have structural patterns that can be mapped into code functions similar to how math problems can mapped into code functions. A key point of this paper is that this method is task agnostic as compared to previous methods that manually create datasets for a specific domain i.e. legal domain when wanting to reduce hallucination.

Because there's no ground truth for this kindve data the authors then come up with a generate pseudo-labels to supervise model training and have some filtering checks to ensure they get high quality data. They create their dataset from Wikipedia and evaluate on Q/A and summarization tasks. They finetune three LLMs: Llama, Mistral and Qwen and find that their method reduces hallucination and improves on accuracy.

The contributions as I see it are
1) The text-to-code and code-to-text training paradigm
2) The methodology used to create this data

**Questions:**

Questions
1) During the fact filtering stage what is considered an invalid sentence? I understand factual / non-factual sentence.

2) Is ROUGE a strong enough metric for semantic fidelity versus using model-based metrics?

3) It was mentioned that "For any code that fails the syntactic validity check, we assign a fixed similarity score of Si = 0.1". Why not just throw the data away?

Suggestions
1) I suggest you put the fact filtering performance that is listed in the appendix into the main text as that was a big question I had.

2) There were some typos of Mistral vs Ministrel

**Ethical Concerns:**

["NO or VERY MINOR ethics concerns only"]

**Final Justification:**

My score was already a Strong Accept and the authors addressed any small questions I had.

**Limitations:**

yes

**Quality:**

4

**Strengths And Weaknesses:**

Strengths

1) I think the paper is well motivated and the idea to map factual statements into the coding paradigm is very interesting to my knowledge novel.

2) The experiments are well thought out and it seems the authors compare against the latest methods SymbCoT and Lookback.

3) The fact that this method is task agnostic is great and if the dataset gets released publicly I think will be useful for the community

Weaknesses

My overall concern with the paper is that there are parts that aren't clear but I think can be easily addressed. More specifically:

1) The pseudo-labeling mechanism is a little unclear. Do you generate the data starting from the base model then train it and regenerate the data to improve it's quality or is it all in one pass?

2) In Figure 4 it seems w/o Code2Text is the worst performing model but in the text in Line 252-253 you mention retaining only the
code-to-text pathway results in the largest increase in hallucination rate which seem contradictory. Also I would like to know why is this the case versus why text-to-code is more helpful.

3) In Table 1 it seems SymbCoT and Lookback aren't as good as your more simpler baselines (SFT vs prompt) does this mean they are not strong enough baselines to compare against?

4)  It was a little hard to follow the section on Semantic Fidelity via Reverse Reconstruction. Like what do we mean by reorganized template and substituting runtime values.

5) It was a little unclear if after your training the inference takes in text as input and text as output or if it was a two pass inference where you first ask the model to generate the code then the text. Section Effectiveness of Structured Code-based Inference cleared it up but that part came later and caused some confusion.

---

> ### Author Rebuttal · Authors · 2025-07-31
>
> We are truly grateful for your constructive feedback and for acknowledging our work. Below, we offer comprehensive answers to your concerns.
>
> **1.Invalid Results in Fact Filtering Stage:** Thank you for your insightful question. In Appendix B.1 of the paper, we reported the distributions and classification performance of different text types, but did not explain the invalid results in detail. We apologize for this oversight and any resulting confusion. An invalid text refers to any input for which the LLM does not provide a clear “yes” or “no” label. To evaluate the classification accuracy for invalid text types, we manually reviewed 100 cases and found that 96% truly lacked valid labels. This typically occurs when the model fails to follow instructions, outputs something in an unexpected format, or cannot make a judgment due to the complexity or ambiguity of the input. We will add these details in the revised appendix.
>
> **2.Evaluation of Semantic Fidelity:** Thank you for raising this important point. We chose the combination of ROUGE-1 and ROUGE-L as our semantic fidelity metrics due to their complementary strengths: ROUGE-1 captures factual entities, while ROUGE-L reflects word order and structural similarity—both crucial for evaluating semantic and contextual relationships.
>
> In addition to $S_i$ Score (ROUGE-based metric as defined in Eq. 7 of the main paper), we evaluated the quality of generated code across training iterations using BERTScore, which measures deeper semantic similarity using contextual embeddings. The evaluation results indicate that  $S_i$ and BERTScore remain closely aligned and both improve across training rounds are presented below.
>
> |  Iteration  | $S_i$  Score (%) | BERTScore (%) |
> |:-----------:|:-----------:|:-------------:|
> |  Round 1    |   87.96     |     88.45     |
> |  Round 2    |   91.61     |     91.82     |
> |  Round 3    |   92.71     |     92.50     |
>
> To further assess the logical equivalence between reconstructed text based on generated code and original texts, we conducted a deeper evaluation using DeepSeek-R1. After the third round of iterative training, the data were split into High-Quality (Si > 0.8) and Low-Quality (Si ≤ 0.8) subsets. Alignment rates for (1) entity overlap and (2) relationship overlap were calculated with both DeepSeek-R1 extraction and manual counting. Manual verification of 100 random samples showed DeepSeek-R1 achieves 92% overall accuracy. The evaluation results are as follows:
>
> |   Dataset    | Entity Alignment Rate (%) | Relationship Alignment Rate (%) |
> |:------------:|:----------------------------------:|:-----------------------------:|
> | **High-Quality** |              **92.35**                 |            **86.26**              |
> | Low-Quality  |              54.33                 |            38.16              |
>
> The High-Quality set achieved much higher factual alignment, indicating that alternating code-text training helps the model capture logical relationships. These results also support that ROUGE-based evaluation, though simple, aligns well with more rigorous metrics and serves as a practical proxy for semantic fidelity.
>
> **3.Handling of Syntactically Invalid Code:** Thank you for your thoughtful question. We chose not to discard syntactically invalid code because our training process is iterative and self-improving. Assigning a fixed low score ensures these challenging cases receive greater loss and remain a focus for the model to improve upon. As training progresses and the model’s code generation ability improves, some previously invalid or low-quality cases may become correctable. By penalizing rather than discarding such cases, the model has repeated opportunities to learn from these difficult examples throughout training.
>
> **4. Clarification on Pseudo-label Generation and Inference Procedure:** Thank you for your question regarding pseudo-labeling. To clarify, we adopt an iterative pseudo-labeling process: in each round of Alternating Code-Text Training, we regenerate pseudo-code labels using the current model checkpoint and use them as supervision for the next round. As alternating training progresses, the model’s representations of text and code become increasingly aligned, which continuously improves its text-to-code generation ability. Consequently, the quality of the regenerated pseudo-labels also increases over time. These higher-quality pseudo-labels provide more effective supervision in subsequent rounds, further enhancing the model’s performance and modality alignment. This mutually reinforcing process allows the model to benefit from both improved alignment and better supervision as training advances. We will update the manuscript to clearly describe this process.
>
> Regarding the inference procedure, our main approach operates in a standard text-to-text manner, where the model directly generates the output text from the input text without an intermediate code generation step. The "Effectiveness of Structured Code-based Inference" section presents an additional two-stage inference experiment to assess the utility and reliability of code generated from text as an intermediate representation; however, this is not the primary inference setup used throughout the paper.
>
> **5.Explanation of Figure 4 Ablations and Pathway Importance:** Thank you for pointing out the confusion. We apologize for any confusion caused by the figure legend. In Figure 4, “w/o Alternating (Code2Text)” actually means we retain only the code-to-text pathway and remove text-to-code training. We realize that the notation may be ambiguous and could be misinterpreted; we will revise the figure legend to clarify that this variant keeps only code-to-text. This is consistent with the main text (Lines 252–253), which states that removing text-to-code (i.e., keeping only code-to-text) leads to the largest increase in hallucination rate, as shown by the poor performance of this variant in Figure 4.
>
> Regarding why text-to-code is more helpful: the text-to-code pathway is particularly valuable because it guides the model to parse and organize factual content into code-like, logically consistent representations, thereby promoting stronger alignment between natural language and structured information. This process helps the model learn more consistent contextual logic and improves factual reliability. Meanwhile, the code-to-text pathway primarily converts structured code back into natural language, supporting the model’s ability to generate fluent and semantically coherent text. While both pathways contribute to overall performance, our results suggest that text-to-code is especially effective in mitigating hallucinations.
>
> **6.Discussion of SymbCoT and Lookback Baselines:**  Thank you for your helpful comment. SymbCoT is a strong baseline recently proposed in 2024, which is based on symbolic reasoning and logic rules and is particularly effective for tasks involving first-order logic and constraint optimization. However, our summarization and QA datasets contain not only problems that can be formulated as first-order logic, but also more complex or open-ended logical relationships. SymbCoT is less capable of handling these complex cases, which may explain its performance degradation on our benchmarks.
>
> Lookback is another recent baseline that detects and mitigates input-conflicting hallucinations by training a classifier on attention-based features, typically requiring 1k–2k hallucination-annotated examples for effective training. According to the original paper, these annotations are generated automatically using GPT-4o. Our manual review of 100 SQuAD V2 samples showed that the accuracy of GPT-4o-based annotation was only 82%. We speculate that this may have contributed to the suboptimal performance of the method.
>
> Prompt-based and SFT methods are directly based on LLMs and are less constrained by symbolic expressivity or the need for hallucination-specific annotations. They can better leverage the generalization and flexibility of large language models, resulting in more robust performance across diverse tasks. We will add these analyses to the revised manuscript.
>
> **7.Reorganized Templates and Value Substitution:** Thank you for your question. In the "Semantic Fidelity via Reverse Reconstruction" section, a reorganized template refers to a text template generated by the model, where specific information such as entity names, attributes, and relationships are replaced with code variables or placeholders (e.g., mitra_ghosh_publishers.founders[0].first_name). This intermediate template makes explicit which parts of the text are linked to code entities.
>
> Substituting runtime values means that when we execute the generated code, each placeholder variable in the reorganized template is automatically replaced with its actual value from the data at runtime. For example, the placeholder mitra_ghosh_publishers.founders[0].first_name is filled in with "Mitra", resulting in a fully instantiated text.
> This process ensures that the reconstructed text (after substitution) can be directly compared to the original text for semantic fidelity. By doing so, we can systematically evaluate whether the generated code truly preserves the detailed meaning and factual content of the original text.
>
> **8.Placement of Filtering Results & Typographical Errors:** Thank you for your valuable suggestions. We agree that presenting the fact filtering results in the main text will improve clarity, and we will move these results accordingly in the revised manuscript. Additionally, we appreciate you bringing the typographical errors to our attention; we will thoroughly review the manuscript to ensure all such errors are corrected.

---

> > ### Comment · Reviewer_mHXd · 2025-08-06
> >
> > Thank you for answering my questions in detail. My questions have been resolved.

---

> > > ### Author Response · Authors · 2025-08-06
> > >
> > > We sincerely thank you for the time and effort you have dedicated to reviewing our paper. Your thoughtful comments and questions were very valuable and helped us further complete our work.

---

### Official Review · Reviewer_Hran · 2025-07-02

**Clarity:** 3
**Significance:** 3
**Originality:** 3
**Rating:** 5
**Confidence:** 4

**Summary:**

The paper proposes a Fact-driven Alternating Code-text Training approach (called FACT) to mitigate the hallucinations generated by the LLMs. The approach relies on the grounding that the intrinsic structural patterns in texts can directly be mapped into programmable constructs. The training regime includes both text-to-code and code-to-text predictions by the LLMs. In the absence of direct training supervision, the authors employ a pseudo-labeling technique that ensures syntactic correctness and semantic faithfulness of the original texts. The empirical studies carried out in the paper shows in the improvements in consistency even when trained with a smaller training set. The paper also demonstrates the versatility of the approach in terms of increased performance in summarization and QA tasks without any task supervision.

**Questions:**

1. The authors should mention the selection criteria of the LLMs.
2. The formulation of S_i should be justified with appropriate reasons.
3. The authors should give links to appropriate sections in Appendix (use \ref in latex).
4. The authors should add a section in the appendix detailing the evaluation metrics.

**Ethical Concerns:**

["NO or VERY MINOR ethics concerns only"]

**Final Justification:**

I have reviewed the author rebuttal and they have answered the question regarding S_i and the choice of LLMs, and details about the evaluation metrics. This resulted in increase in my rating from 4 to 5.

**Limitations:**

Yes

**Quality:**

3

**Strengths And Weaknesses:**

The paper presents a task agnostic framework for mitigating hallucinations generated by LLMs through alternating code-text training. The current approach acts as a bridge between different modalities of inputs where unambiguous and clear representations can be obtained. The authors also minimize the extent of direct training supervision that exhibits emergent capabilities in unseen tasks. The authors make use of pseudo-labels for assisting in training supervision ensuring syntactic correctness and semantic faithfulness of the original texts. The paper attempts to address input-conflicting and context-conflicting hallucinations that are the most prevalent types of hallucinations generated by current LLMs.

The authors do not mention the selection criteria for the baseline LLMs. It is not clear why LLMs with lower number of parameters are chosen. The formulation of S_i lacks details and intuition. It is unclear why ROUGE-1 and ROUGE-L are combined, and not other variants of ROUGE. The details of the evaluation metrics are missing. The authors should give links to appropriate sections in Appendix. In the current version, it is missing. There are a few grammatical errors in the paper that needs thorough proof-read.

---

> ### Author Rebuttal · Authors · 2025-07-31
>
> We greatly appreciate your thoughtful feedback and your acknowledgement of our work. Our detailed responses to your concerns are provided below.
>
> **1.Selection Criteria for Baseline LLMs:** We appreciate your suggestion and agree that clarifying the selection criteria for the baseline LLMs will improve the transparency of our work. We selected LLaMA-3.1-Instruct-8B, Ministral-Instruct-8B, and Qwen-2.5-Instruct-7B as base models because they are recent and widely used instruction-tuned LLMs that demonstrate strong performance on a variety of NLP tasks. Additionally, we chose models with similar parameter sizes (7B/8B) to ensure a fair and direct comparison under comparable computational and resource constraints. These models are all publicly available, making them easily accessible for the research community. We will add these selection criteria to the revised manuscript.
>
> **2.Motivation and Details of $S_i$:** Thank you for your constructive feedback. In our alternating training scheme, the text-to-code SFT stage relies on model-generated code as pseudo-labels, whose quality is critical. To ensure reliable supervision, we introduce the $S_i$ metric as a unified and fine-grained measure of code quality, integrating both syntactic validity and semantic fidelity. By weighting the training loss with $S_i$, the model is dynamically penalized for lower-quality generations, encouraging it to focus on the most challenging cases and improving robustness in the absence of ground-truth code.
>
> Choice of ROUGE-1 and ROUGE-L: We chose the combination of ROUGE-1 and ROUGE-L as our semantic fidelity metrics due to their complementary strengths: ROUGE-1 captures factual tokens and entities, while ROUGE-L reflects global word order and structural similarity—both crucial for evaluating semantic and contextual relationships. To validate this choice, we conducted hallucination evaluation experiments on the CNN/Daily Mail dataset using a Llama-based model, comparing different ROUGE metrics and their combinations. As shown in the table below, ROUGE-1 and ROUGE-L, either individually or combined, consistently outperformed other variants, and adding ROUGE-2 did not provide further improvements.
>
> | Similarity Metric  | Consistency | AlignScore |
> |-----------------------|-------------|------------|
> | ROUGE-1 only  | 89.33       | 88.25      |
> | ROUGE-2 only   | 88.11       | 86.04      |
> | ROUGE-3 only   | 87.11       | 86.12      |
> | ROUGE-L only   | 88.42       | 86.90      |
> | **ROUGE-1 + ROUGE-L (Ours)**   | **91.08**   | **87.15**  |
> | ROUGE-1 + ROUGE-2 + ROUGE-L  | 91.03       | 86.44      |
>
> The setting of the $S_i$ hyperparameter: During the initial development of our method, we conducted a hyperparameter study to select the optimal penalty value for $S_i$ when the generated code is not executable. We compared several candidate values ($S_i = 0.01, 0.1, 0.2, 0.3$), and for each, evaluated (1) the executable code ratio over three rounds of alternating training, and (2) the final consistency metric (hallucination rate) on the CNN/Daily Mail dataset using a Llama-based model. The results are shown below:
>
> |   Penalty $S_i$   | Executable Code Ratio (1st) | Executable Code Ratio (2nd) | Executable Code Ratio (3rd) | Consistency (%) |
> |:-------------:|:--------------------------:|:--------------------------:|:--------------------------:|:---------------:|
> |     0.01      |          84.33%            |          85.77%            |          85.26%            |     87.42%      |
> |     **0.1**       |          **87.28%**            |          **88.50%**            |         **88.30%**            |     **91.08%**      |
> |     0.2       |          86.40%            |          87.30%            |          87.62%            |     89.24%      |
> |     0.3       |          85.02%            |          85.50%            |          85.14%            |     88.01%      |
>
> As shown in the table, $S_i=0.1$ achieves the best trade-off in both executable code ratio and consistency. Smaller values (e.g., $S_i=0.01$) lead to lower and less stable performance, while larger values degrade results. Thus, we set $S_i=0.1$ based on optimal empirical outcomes. We will add these details to the appendix of the revised manuscript.
>
> **3.Evaluation Metrics Details:**  Thank you for your helpful suggestion. While we have provided references for all metrics, we agree that detailed explanations are needed, especially since some metrics are quite recent (e.g., AlignScore from 2023 and Anah-v2 from 2024). In the revised manuscript, we will add a dedicated appendix section describing all evaluation metrics used in our study, including hallucination, summarization, and QA metrics.
>
> **4.Appendix Links and Grammar Corrections:**  Thank you for your valuable suggestions. We agree that clickable links to specific appendix sections can greatly improve the reader experience. In the revised manuscript, we will update all appendix references in the main text to use the LaTeX \ref command, ensuring that readers can directly access the relevant sections with a single click. Additionally, we will carefully proofread the manuscript to correct all grammatical errors and improve the overall quality of the writing.

---

> ### Author Response · Authors · 2025-08-06
>
> We sincerely appreciate your constructive suggestions. We will incorporate these valuable responses into the revised manuscript.

---

### Official Review · Reviewer_bv6X · 2025-07-03

**Clarity:** 3
**Significance:** 3
**Originality:** 3
**Rating:** 5
**Confidence:** 3

**Summary:**

The paper proposes a framework called Fact-driven Alternating Code-text Training (FACT) to address inconsistent hallucinations in large language models. The paper claims that there are two types of errors: input conflicting and context conflicting. To tackle this, the FACT framework trains the LLM by alternating two tasks, which are text-to-code and code-to-text. The core idea behind this is that factual text has an inherent structure that can be mapped to code. The experiment results show that such training method could reduce inconsistent hallucinations and improve overall performance on QA and summarization tasks.

**Questions:**

1. How will this training method affect the model's other capabilities? Will there be performance drop?
2. How will the system handle very complex factual relationships?
3. How to make sure that the model can convert code back into text faithfully?

**Ethical Concerns:**

["NO or VERY MINOR ethics concerns only"]

**Final Justification:**

The rebuttal from the authors has addressed most of my concerns.

**Limitations:**

yes

**Quality:**

3

**Strengths And Weaknesses:**

## Strengths
1. The paper proposes a novel framework that can reduce inconsistent hallucinations across a wide range of tasks.
2. The experiment shows that the proposed FACT framework is effective, with clear hallucination reduction and task performance improvement.
3. The method seems to be general as it works on 3 different large language models (Llama, Mistral and Qwen)

## Weaknesses
1. The FACT framework will likely work for clear and unambiguous facts. There could be corner cases that makes it hard to map it to the code. There could be ambiguity and figurative text.
2. The method is hard to scale up. The training efficiency will be lowered when there are extra steps like defining a code-like structure, converting text to code and then evaluate their factual consistency.
3. The tested datasets are a bit out dated. Newer and larger datasets are helpful to evaluate the effectiveness of the method (because the base models are relatively newer)

---

> ### Author Rebuttal · Authors · 2025-07-31
>
> We sincerely appreciate your kind feedback as well as your recognition of our work. Below are our responses to each of the concerns you raised.
>
> **1. Impact on the Model’s Other Capabilities:** Thank you for your question. Our method adopts an alternating training scheme between text-to-code and code-to-text, with each direction implemented using supervised fine-tuning training (SFT)—a widely used approach that is generally not found to significantly impair model’s other abilities.
>
> To address this concern, we evaluated not only hallucination metrics but also core task metrics such as Coherence and Relevance for summarization, and F1/EM for QA. As shown in Table 2 of the original paper, our approach even outperforms the base model on these metrics (+6.35% on average), indicating not only no performance drop but also an overall improvement. We believe this improvement is partly due to reducing the "hallucination snowballing" effect [1], where error accumulation degrades output quality. By mitigating such errors, FACT promotes both more accurate generation and better overall performance.
>
> Beyond these results, we acknowledge the importance of broader evaluation for model’s other capabilities and plan to further investigate this in future work.
>
> > [1] Muru Zhang, Ofir Press, William Merrill, Alisa Liu, and Noah A Smith. How language model 442 hallucinations can snowball. In Forty-first International Conference on Machine Learning.
>
> **2.Discussion on complex factual relationships:** For very complex factual relationships, our system decomposes them into multiple simpler components wherever possible, each represented as discrete objects, properties, or functions in the code.
>
> To empirically evaluate this capability, we used code-based filtering and manual confirmation to select 100 samples with complex multi-entity relationships from the Wiki-40B-en dataset (e.g., indirect geographic descriptions and intricate character relationships). Using our trained LLaMA-based model, we generated code for each sample and evaluated executability and text reconstruction fidelity $S_i$ as described in the main paper. We also assessed deeper semantic fidelity with BERTScore. For $S_i$ and BERTScore, we report both the mean and standard deviation (shown in parentheses). The evaluation results are as follows:
>
> | Metric                        | Value   |
> |-------------------------------|---------|
> | Executability Rate            | 0.82  |
> | Semantic Similarity ($S_i$)   | 0.83 (0.17)  |
> | BERTScore                     | 0.81 (0.21) |
>
> These results indicate that our method can handle a range of complex factual relationships. Nevertheless, we acknowledge that extremely intricate may remain challenging and could require more advanced schema designs in future work.
>
> **3.Faithfulness of Code-to-Text Generation:** Our model is alternately trained in both the text-to-code and code-to-text directions using supervised learning, with ground-truth text serving as direct supervision for code-to-text generation. The loss is computed by comparing the generated text with the original reference, thereby encouraging faithful reconstruction and optimizing model performance.
>
> To further assess model fidelity, we evaluated texts generated from code after the third training round on the LLaMA-based model using ROUGE-L and BERTScore, two widely adopted metrics for structural and semantic similarity. As shown in the table below, these results indicate that the model demonstrates robust faithfulness in Code-to-Text generation.
>
> |   Metric    | Average | Standard Deviation | Median |
> |:-----------:|:-------:|:------------------:|:------:|
> |  ROUGE-L    |  0.93   |       0.14         |  0.89  |
> | BERTScore   |  0.92   |       0.17         |  0.90  |
>
> **4. Discussion on Corner Cases:**  Thank you for your thoughtful review. We agree that the FACT framework is most effective for clear, fact-based texts, and that its capacity to handle ambiguous or figurative language is currently limited. Our main goal is to enhance logical consistency and reduce hallucinations in downstream tasks that mainly rely on factual reasoning. This focus is driven by the demands of real-world LLM applications—such as healthcare and legal domains—where the reliability of factual information is crucial. Addressing ambiguity and figurative language remains an open challenge and is a promising direction for future work.
>
> **5. Discussion on Training Efficiency:** Thank you for your valuable comment. We would like to clarify that the code-like structure is not an additional or repeatedly defined step, but is directly integrated into the supervised training process through a fixed prompt. This prompt is constructed once and reused throughout training, eliminating any need for dynamic or manual intervention.
>
> Although our method involves steps such as generating pseudo-labels and evaluating the syntactic and semantic quality of code outputs (since no code ground truth is available), these procedures are fully automated using tools like Python interpreters and LLMs. Our statistics show that training with our method is approximately 1.5 times slower than standard supervised fine-tuning (SFT), representing a modest but manageable overhead.
>
> Moreover, our results demonstrate that FACT achieves substantial improvements on downstream QA and summarization tasks, even with a small amount of general fact-driven data for training. While larger-scale experiments could provide further insights, they are left for future work.
>
> **6. Dataset Selection:** Thank you for your valuable feedback. In our experiments, we deliberately included both established and recent benchmark datasets for a comprehensive evaluation. For the QA task, we used SQuAD v2 (a widely recognized classic) and HaluEval, a recently introduced dataset in 2023 that is specifically designed to evaluate hallucinations in LLMs. For summarization, we included both CNN/Daily Mail and SAMSum; although SAMSum was released in 2019, it is a high-quality, human-annotated benchmark for dialogue summarization. We agree that incorporating even newer and larger datasets could further strengthen the evaluation and will consider this in future work.

---

> > ### Comment · Reviewer_bv6X · 2025-08-05
> >
> > I thank the authors for their response! Most of my concerns are addressed and I'll raise the score to accept.

---

> > > ### Author Response · Authors · 2025-08-06
> > >
> > > Thank you very much for your time and effort in reviewing our work. We truly appreciate your insightful comments and suggestions, which have helped us improve and complete our work.

---

### Decision · Program_Chairs · 2025-09-17

**Decision:**

Accept (poster)

**Comment:**

The paper proposes FACT, a fact-driven alternating code-text training framework to reduce hallucinations in LLMs by mapping factual text into structured code and alternating between text-to-code and code-to-text training. Strengths are the novelty of using code for consistency, task-agnostic design, and solid empirical gains across QA and summarization. Weaknesses include limited theoretical grounding, reliance on pseudo-labeling, narrow task scope, and initially insufficient statistical rigor. The decision to accept is based on the importance of the problem, originality, and consistent improvements across models and datasets, though not strong enough for spotlight/oral. In rebuttal, concerns about robustness, metrics, novelty, and dataset choice were raised; the authors addressed these with multi-seed experiments, statistical tests, deeper analysis of fact filtering, clarifications, and discussion of related work. Most reviewers were satisfied and raised their scores, with one remaining skeptical but moving to weak accept.